# PARALLELPROMPT: Extracting Parallelism from Large Language Model Queries

**Steven Kolawole**
Carnegie Mellon University
skolawol@cs.cmu.edu

**Keshav Santhanam**
Stanford University
keshav2@stanford.edu

**Virginia Smith**
Carnegie Mellon University
smithv@cmu.edu

**Pratiksha Thaker**
Carnegie Mellon University
pthaker@andrew.cmu.edu

## Abstract

LLM serving systems typically treat user prompts as monolithic inputs, optimizing inference through decoding tricks or inter-query batching. However, many real-world prompts contain *latent semantic parallelism*—decomposable structures where subtasks can be executed independently to reduce latency while preserving meaning. We introduce PARALLELPROMPT, the first benchmark for measuring intra-query parallelism in natural user prompts. Our dataset comprises over 37,000 real-world prompts from public LLM chat logs, each annotated with a structured schema capturing task templates, shared context, and iteration inputs. These schemas are extracted using LLM-assisted prompting with rule-based multilingual validation. To evaluate the benefits of decomposition, we provide an execution suite that benchmarks serial vs. parallel strategies, measuring latency, structural adherence, and semantic fidelity. Our results show that intra-query parallelism can be successfully parsed in over 75% of curated datasets, unlocking up to $5\times$ *speedups* on tasks like translation, comprehension, and comparative analysis, with minimal quality degradation. By releasing this benchmark, curation pipeline, and evaluation suite, we provide the first standardized testbed for studying structure-aware execution in LLM serving pipelines.

## 1 Introduction

Large language models (LLMs) are increasingly deployed in interactive systems, powering personal assistants, tutoring agents, and productivity tools. These settings demand both high-quality outputs and low-latency inference. While much progress has been made on inter-query optimizations, e.g., batching requests, model compression, and decoding, a complementary axis remains underexplored: *intra-query parallelism*. **Can we speed up inference by decomposing a single user prompt?**

In this work we focus on natural language prompts that implicitly contain multiple independent subtasks, which we refer to as *latent semantic parallelism*. For example, asking a model to generate ten short stories will take longer than executing ten generation calls in parallel (see examples of such prompts from public LLM logs in Figure 1). To study this phenomenon at scale, we develop PARALLELPROMPT, a benchmark of over 37,000 real-world prompts drawn from LMSYS-Chat-1M [1] and WildChat-1M [2]. Our multi-stage pipeline (overview in Figure 2) filters these logs for parallelizable prompts, yielding a surprising 10% of user queries with decomposable structure. This validation rate represents a conservative, high-precision estimate that translates to practical impact: in production deployments handling billions of monthly queries [3, 4, 5], this yields millions of optimization opportunities with latency improvements. For context, even widely adopted optimizations like speculative decoding [6, 7] and KV-caching [8] only benefit certain query types, yet are considered valuable

```
Generate 10 variations of detailed
descriptions of a room, describing the
type of room, the style, and the included
furniture. The description is based on the
following list: ["bed", "table",
    "nightstand", "lamp", "mirror"]
```

**(a)**

```
How acceptable are the following
English sentences on a scale of 1 to 10?

1. The book is brown.
2. The book are brown.
...
```

**(b)**

**Figure 1:** Examples of real user prompts with latent parallel structure. *(a)* Example of a repeated-generation query, where the 10 generations can be executed in parallel. *(b)* Example of a classification query, where the task (rating sentences) can be parallelized across the queries (each sentence).

production techniques. Perhaps more importantly, intra-query parallelism complements rather than competes with existing optimizations, offering further gains within individual queries.

Although prior methods have explored related problems in task decomposition, e.g., Skeleton-of-Thought [9], Tree-of-Problems [10], and LLMCompiler [11], these approaches target synthetic scenarios or predefined schemas. Interestingly, we find that prior task decomposition methods fail on many of the prompts in our benchmark. This limitation stems from fundamental architectural constraints: prior methods make strong assumptions about task structure that don't generalize to the diverse parallelization patterns present in real user prompts. In contrast, our addresses naturally occurring user prompts where parallelizable structure must be discovered rather than assumed.

PARALLELPROMPT enables both method and system evaluation. Our results show that naive serial execution is often suboptimal, with simple parallelization yielding $3\times$–$5\times$ latency improvements on tasks like translation and comprehension. We also surface failure cases like dependency blindness, highlighting open challenges for future work. Overall, we make the following contributions:

- We introduce *intra-query semantic parallelism* as a new axis for LLM efficiency, complementing token- and batch-level optimizations.

- We release PARALLELPROMPT, the first benchmark of real user prompts with decomposable structure. Our benchmark includes a taxonomy of decomposition patterns, a multilingual schema extraction pipeline, and an execution suite measuring latency, structural adherence, and semantic fidelity. We provide open-source data and reference implementations to support experimentation.

- Using our benchmark, we evaluate recent decomposition methods that could be applied to intra-query parallelism, and find that these methods fail on many of the practical queries in the broader set of real-world prompts in our benchmark. Our analyses also reveal other interesting failure cases and characteristics of real-world parallelizable prompts—revealing a number of open problems and highlighting the importance of the benchmark for future work in this area.

## 2 Related Work

Optimizing LLM inference efficiency has been a central focus in systems and NLP communities. While most work targets decoding acceleration or model compression, a growing body explores restructuring tasks to improve end-to-end latency. We organize prior work into: (1) prompt decomposition for tool or symbolic execution, (2) serving-time parallelism, and (3) evaluation benchmarks.

**Prompt Decomposition.** A number of general frameworks consider decomposing prompts for more structured execution. LLMCompiler [11, 12] reformulates prompts into tool-augmented DAGs but assumes predefined schemas and symbolic APIs. Tree-of-Problems (ToP) [10] decomposes math and logic tasks but relies on compositional structures rarely found in everyday prompts. Skeleton-of-Thought (SoT) [9] parallelizes outline expansions but is limited to tasks with clearly separable parts. Super JSON Mode [13] requires predefined output schemas for structured parsing.

In contrast, PARALLELPROMPT targets *naturally occurring, unstructured prompts* with latent parallelism. We induce schemas from real-world prompts using LLM-assisted extraction and evaluate execution tradeoffs in latency and semantic fidelity. Unlike prior works that rely on hardcoded templates or task-specific logic, our benchmark spans diverse, real-world prompt types—including emerging categories not explicitly seen during schema design (Section 3; see Appendix C.2 for full analysis). To assess how these methods generalize beyond their original settings, we include evaluations of ToP and SoT on our curated benchmark as representative decomposition baselines, finding that these approaches fail on a number of the real-world prompts in our benchmark.

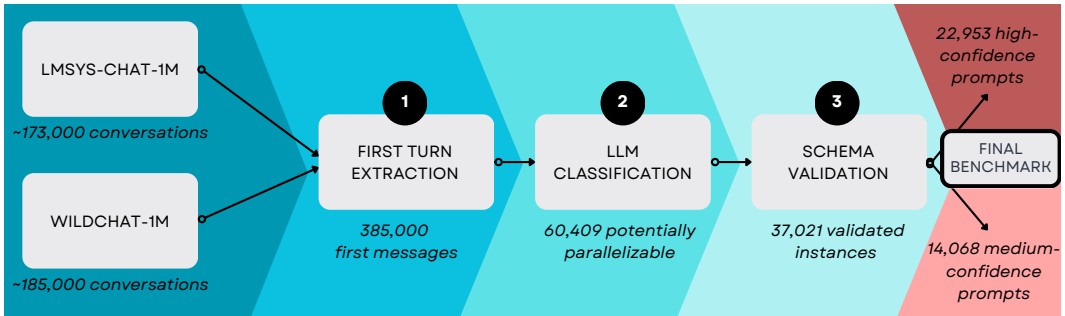

**Figure 2: PARALLELPROMPT curation pipeline.** Our multi-stage filtering process identifies naturally occurring parallelizable structures in real-world LLM interactions. Surprisingly, over 10% of user prompts contain latent parallel structure. High-confidence instances contain explicit structural markers like numbered lists or item delimiters, while medium-confidence rely on semantic cues such as plural forms or task multiplicity. This precision-focused validation ensures benchmark quality for measuring intra-query parallelism benefits.

**Serving-Time Parallelism.** Batch-level systems like vLLM [14] and TensorRT-LLM [15] improve throughput by executing multiple queries concurrently but treat each prompt as atomic. Token-level methods like Speculative Decoding [6, 7], Lookahead Decoding [16], and Medusa [17] reduce autoregressive bottlenecks by predicting multiple tokens in parallel. Other methods include non-autoregressive generation [18], blockwise decoding [19], and attention stabilization [20]. These techniques optimize serving pipelines but do not exploit **semantic parallelism**—restructuring a *single* prompt into independent, meaningful subtasks. Our work complements these methods by surfacing *latent intra-query parallelism* through learned schema extraction and evaluation. Frameworks such as DSPy [21], SGLang [22] and LangChain [23] enable parallel execution of downstream tasks generated from a single task, but also do not aim to discover execution structure within user queries.

**Benchmarks and Datasets.** Most benchmarks that evaluate structured queries focus on synthetic or expert-curated tasks. BIG-Bench [24] and BIG-Bench Hard [25] evaluate structured reasoning on expert-designed tasks. GSM8K [26] targets math reasoning with chain-of-thought. WizardLM [27] generates synthetic instructions but lacks grounding in real user prompts. HotpotQA [28] covers multi-hop QA but focuses on factoid questions, not open-ended instructions. In contrast, PARALLEL-PROMPT is grounded in *naturally occurring* user interactions from large-scale LLM logs. We annotate latent decomposable patterns and evaluate execution across latency, fidelity, and generalization—providing a realistic testbed beyond synthetic or templated tasks.

## 3 The PARALLELPROMPT Benchmark

PARALLELPROMPT tackles a foundational but overlooked question: *when is it possible to parallelize within a single user prompt?* While LLM serving pipelines typically optimize across independent requests, many real-world prompts themselves contain latent decomposable structure that, if identified, can unlock substantial latency and throughput gains. PARALLELPROMPT is the first benchmark to systematically capture and evaluate this form of intra-query parallelism at scale.

We build PARALLELPROMPT by curating over 37,000 naturally occurring prompts from public LLM usage logs, each annotated with structured schemas that reveal how the prompt can be partitioned into semantically independent subtasks. These annotations enable reproducible evaluation across diverse decomposition patterns, task types, and execution strategies, providing a realistic testbed for parallelization research.

### 3.1 Benchmark Design and Curation

**Data Sourcing and Filtering.** Our benchmark is grounded in real-world interactions sourced from WildChat-1M[1] and LMSYS-chat-1M[2], two of the largest publicly available LLM chat datasets. We

---

[1]WildChat-1M is released under the Open Data Commons Attribution License (ODC-By). See https://opendatacommons.org/licenses/by/1-0/.

[2]LMSYS-Chat-1M carries a more restrictive license; however, we obtained explicit permission from the authors (April 2025) to release our curated subset for the benchmark.

extract the first user message from each conversation, yielding over 2 million standalone prompts spanning casual queries, instructional tasks, and multi-step requests. As shown in Figure 2, we employ a multi-stage pipeline that first classifies prompts using Claude 3.5, then extracts structured schemas capturing task templates, context, and iterable data elements. Crucially, our system prompts the model to *generate full schemas*, not just binary classifications. This design choice surfaces both decomposition structure and reliability, enabling robust downstream validation (Appendix A.1).

Our validation pipeline explicitly prioritizes precision over recall, where precision represents the fraction of validated prompts that can be executed in parallel without semantic degradation, and recall represents the fraction of all inherently parallelizable prompts that our pipeline successfully identifies. This conservative approach ensures benchmark quality while acknowledging that true parallelism rates could be higher with improved extraction methods and more so for specialized domains.

Each validated prompt is annotated with a structured schema specifying its decomposable components, including a task template with placeholders, shared context, and either a list of items or a generation count. The schemas follow a consistent five-field format detailed in Appendix A.2, ensuring reproducible decomposition and execution.

**Category Taxonomy and Structural Diversity.** As shown in Figure 3, our dataset spans seven canonical categories: Repeated Generation (25%), Reading Comprehension (30%), Named Entity Recognition (30%), Keyword Extraction (7%), Translation (9%), Language Correction (6%), and Sentiment Analysis (3%). These account for approximately 95% of validated prompts and represent the most common parallelizable patterns in LLM interactions. The seven canonical categories emerged from extensive manual curation of over 10,000 prompts from LMSYS-Chat-1M during early development, revealing recurring parallelizable patterns that motivated our schema definition before turning to automated extraction with Claude 3.5 for final dataset curation.

Beyond these canonical types, our extraction process surfaced over 400 novel categories that reflect the rich structural diversity of real-world LLM usage. We organize these into six meta-categories (Comparative, Generative, Analytical, Transformative, Organizational, and Computational) detailed in App. C.3, providing a comprehensive taxonomy of parallelizable patterns.

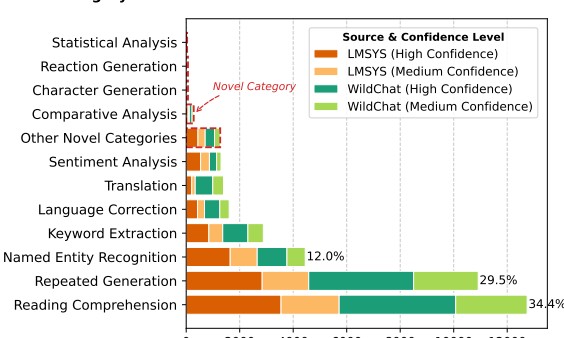

**Figure 3: Distribution of validated categories by source and validation confidence.** The canonical categories dominate (esp. Repeated Generation and Reading Comprehension) since we explicitly optimize for curation of known categories, but the dataset also includes hundreds of emerging, structurally diverse novel categories (e.g., Comparative Analysis, Character Generation). This breadth highlights the benchmark's coverage of both common and specialized parallelization patterns.

**Multilingual Coverage and Extraction Challenges.** PARALLELPROMPT captures parallelizable prompts across at least 11 languages. While English dominates ($\approx$84%), Figure 9 (in the appendix) reveals substantial representation in Chinese (5.6%), Russian (6.3%), and other languages. Importantly, validation success rates vary significantly across languages: European languages show higher structural reliability (55–63% high-confidence), while East Asian languages like Chinese and Japanese exhibit lower rates (28–34%). This disparity reflects both linguistic structural differences and extraction method biases, as explored in Appendix E.1.

Rather than presenting artificially balanced data that would obscure real-world deployment challenges, we, however, preserved the authentic distributional patterns of user traffic (84% English) to expose genuine multilingual fragilities that require addressing. This positions PARALLELPROMPT as the first benchmark to systematically analyze multilingual decomposition challenges, providing detailed failure pattern analysis (Appendix E.1) to guide development of language-aware extraction methods.

## 3.2 Validation Methodology and Quality Control

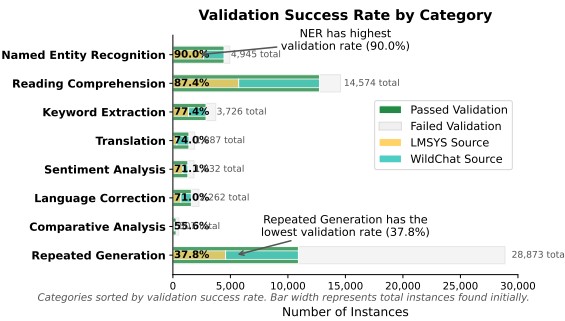

**Figure 4: Validation success rates by category**, showing that structured prompts like Named Entity Recognition and Reading Comprehension pass validation more reliably than creative tasks like Repeated Generation. This pattern highlights the limitations of current schema extraction methods for loosely structured or open-ended prompts.

**Tiered Validation Framework.** To ensure benchmark quality while acknowledging the ambiguity in user-generated prompts, we implement a three-tier validation system (Appendix B.1): high-confidence prompts (62%) featuring explicit structural markers, medium-confidence (38%) relying on softer linguistic cues, and failed validation cases (38.6% of candidates) rejected due to constraint violations.

As Fig. 4 shows, validation success varies significantly by task type. Structured tasks like Named Entity Recognition achieve success rates of 90%, while creative prompts like Repeated Generation pass validation at much lower rates (37.8%). This pattern reveals a tension between creative freedom and structural reliability that impacts parallelization effectiveness.

Our validation approach was grounded in ≈5,500 manual inspections across the three-tier framework (detailed in Appendices B & D.4). This human evaluation revealed that over 75% of structurally validated prompts remain suitable for effective parallelization, confirming the practical applicability of our approach while identifying systematic failure patterns that guide future improvements.

The validation process applies several key constraints to ensure schema integrity, including mutual exclusivity between data and count fields, template-placeholder compatibility, and minimum parallelism thresholds. Common failure patterns (analyzed in Appendix B.3) include template-placeholder mismatches, mutual exclusivity violations between data and count fields, and insufficient parallelism thresholds, providing insights into the limitations of current schema extraction approaches.

### 3.3 Dataset Statistics and Research Implications

**Scale and Prevalence.** From 358,000 inspected prompts, we validated 37,070 as parallelizable (16,721 from LMSYS and 20,349 from WildChat), representing a 10.3% yield. This finding challenges the assumption that intra-query parallelism is rare, revealing instead that it constitutes a significant mode of user interaction with LLMs.

**Structural and Linguistic Patterns.** Our analysis surfaces several notable patterns in the dataset:

- **Category distribution**: Reading Comprehension and Repeated Generation dominate, but the long tail of novel categories expands PARALLELPROMPT's coverage beyond prior benchmarks.
- **Language-specific tendencies**: Chinese prompts disproportionately target Named Entity Recognition tasks (11.33%), while Japanese prompts favor Translation (47.37%).
- **Validation biases**: European languages achieve consistently higher validation rates than East Asian languages, highlighting the need for language-sensitive extraction methods.
- **Novel category characteristics**: Novel categories feature more complex templates (15% longer on average) and higher context utilization compared to canonical categories.

These patterns highlight both the potential and challenges of intra-query parallelism in practice. They also reveal the limitations of current extraction methods, particularly for creative tasks and non-Western languages (Appendix E.3). By combining large-scale data curation, structured schema extraction, multilingual coverage, and tiered validation, PARALLELPROMPT provides the first comprehensive testbed for studying intra-query parallelism in practical settings. Its breadth of prompt types, languages, and structural patterns makes it a valuable resource for advancing structure-aware LLM execution strategies that balance efficiency with output quality.

### 3.4 Benchmark Applications

PARALLELPROMPT enables three primary evaluation scenarios: (1) Schema extraction research, evaluating how well different models or prompting strategies identify parallelizable structure; (2)

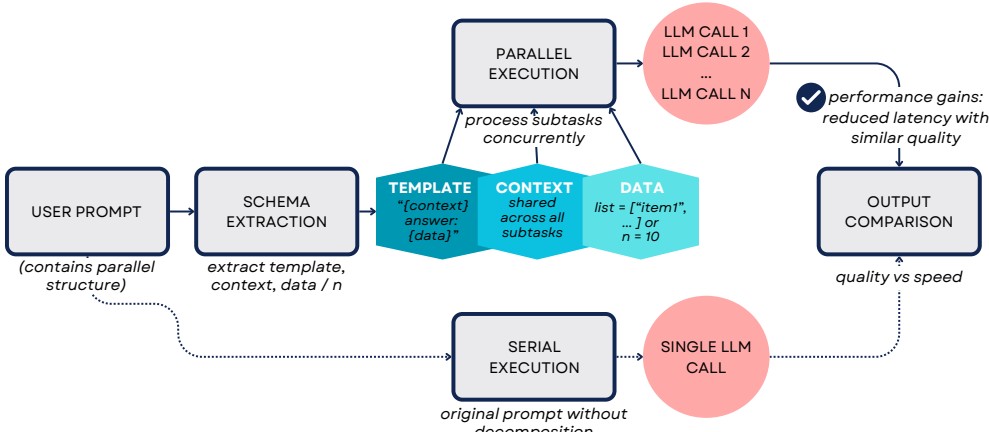

**Figure 5: How PARALLELPROMPT operationalizes intra-query parallelism.** This execution pipeline powers the benchmark's core evaluation, contrasting serial and parallel strategies on real user prompts *regardless of their task category*. By grounding performance measurement in schema-based decomposition and systematic output comparison, the pipeline reveals when parallel execution achieves meaningful speedups without compromising output quality—turning benchmark abstractions into actionable insights for real-world LLM serving systems.

Decomposition execution strategies, through systematic latency-quality tradeoff evaluation across diverse task categories; and (3) Model benchmarking, assessing how different LLMs perform under parallel versus serial execution. The task-agnostic schema format and modular C++ infrastructure support any structure-aware execution pipeline, providing a standardized testbed for advancing structure-aware LLM serving research beyond the specific method evaluated here.

## 4 Evaluating Parallelism in Practice

We next use PARALLELPROMPT to investigate how effectively intra-query parallelism can be leveraged in practice. This section evaluates performance gains, quality implications, and tradeoffs across different task types, providing an empirical foundation for structure-aware LLM serving systems.

### 4.1 Evaluation Methodology

**Category-Agnostic Execution Framework.** As illustrated in Figure 5, our evaluation infrastructure is designed to handle any parallelizable prompt with a valid schema—regardless of its task category or domain. This category-agnostic approach contrasts with prior methods that make strong assumptions about task structure. Our framework requires only a template field, an optional context field, and either a data list or an iteration count to enable parallel execution, making it applicable to both canonical categories and novel patterns discovered in the wild. We implemented this framework as a high-performance C++ backend that provides fine-grained thread control and robust API rate limiting through exponential backoff. This infrastructure measures both latency and output quality across execution strategies, revealing when parallelization yields practical benefits.

Our evaluation pipeline operates in two phases: (1) schema extraction using Claude 3.5, and (2) execution using GPT-4-1106-preview via the OpenAI API. For schema extraction, we evaluated multiple models including Claude-3.7 Sonnet (too powerful, over-extracted with excessive decomposition), GPT-4o-mini (too weak, under-performed with high error rates), and several open-source models via Together.ai API before selecting Claude 3.5 for its optimal precision-coverage tradeoff. Implementation details, including thread management and timeout handling, are provided in Appendix F.1.

**Measurement Framework.** We assess parallelization via three complementary metrics: i) **Latency**, ii) **Semantic Fidelity**–which is the output quality assessed by an independent LLM judge (GPT-4o), and, iii) **Normalized Speedup**, i.e., raw speedup adjusted for output token length differences.

The normalized speedup metric addresses a systematic bias where LLMs processing multiple items sequentially often truncate individual responses due to internal "length budgeting" behavior [29, 30, 31]. For example, requesting one story might yield 500 words, while requesting ten stories results in

**Table 1:** Performance metrics for three primary task categories from PARALLELPROMPT. Measurements exclude schema extraction time except for E2E (end-to-end) which includes full pipeline execution. Normalized speedup accounts for differences in output token lengths between serial and parallel approaches, revealing significant performance gains even after quality adjustment.

| Task Category | Avg Parallel Duration (s) | Avg Serial Duration (s) | Normalized Speedup | Raw Speedup |
|---|---|---|---|---|
| Keyword Extraction | 2.38 | 3.23 | **2.54**× | **1.36**× |
| Reading Comprehension | 3.49 | 10.27 | **5.72**× | **2.94**× |
| Repeated Generation | 3.79 | 9.51 | **4.39**× | **2.50**× |
| Repeated Generation (E2E) | 4.88 | 9.51 | **3.41**× | **1.70**× |

50 words each as the model attempts to fit everything into similar total response lengths. Parallel execution avoids this by allocating full context to each subtask, and our metric adjusts for these length differences to enable fair efficiency comparisons.

For Repeated Generation tasks, we implemented a diversity heuristic that assigns different starting letters to each parallel generation, ensuring outputs remain distinct without relying on shared context. This approach balances generation independence with output diversity, as detailed in Appendix F.3.

**Baseline Comparison.** We evaluate our structured decomposition approach against two established methods: Skeleton-of-Thought (SoT) [9] and Tree-of-Problems (ToP) [10]. While these methods were designed for specialized decomposition patterns, our evaluation examines how they generalize to the diverse parallelization scenarios present in our benchmark. A qualitative comparison of method strengths and limitations is provided in Appendix I.

### 4.2 Performance Results

**Latency Reduction.** As shown in Table 1, parallel execution yields substantial performance improvements across all evaluated categories. Reading Comprehension tasks show the highest raw speedup (5.72×), with Repeated Generation (4.39×) and Keyword Extraction (2.54×) also demonstrating significant gains. Importantly, even when including schema extraction overhead in end-to-end measurements, Repeated Generation tasks still achieve a 3.41× speedup—highlighting the practical value of intra-query parallelism in realistic deployment scenarios.

**Scalability Analysis.** To assess how parallelization benefits scale with task complexity, we conducted an in-depth analysis of Repeated Generation tasks with varying output counts. As Figure 6 illustrates, while serial execution time grows linearly with the number of requested outputs, parallel execution time remains relatively constant. This leads to increasing speedup factors for more complex queries, with some diminishing returns at higher counts due to API rate limiting. This pattern confirms that parallelization becomes increasingly valuable as query complexity grows—precisely when latency improvements matter most.

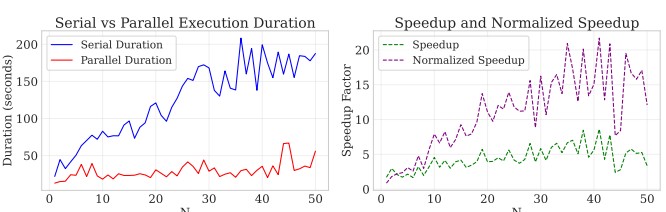

**Figure 6: Latency scaling with number of requested outputs ($n$).** (a) Serial execution time grows linearly with n while parallel time remains relatively constant, demonstrating the core efficiency benefit of intra-query parallelism. (b) Speedup increases nearly linearly with $n$ until API rate limiting becomes a factor, with normalized speedup accounting for quality differences. This scaling pattern suggests parallelization becomes increasingly valuable for complex, multi-part queries.

**Comparative Method Performance.** Our quantitative comparison with SoT and ToP, detailed in Appendix I, reveals the limitations of specialized decomposition approaches when faced with diverse real-world prompts. SoT's two-phase outline-then-expand approach makes assumptions about task structure that don't generalize to extraction tasks, while ToP's recursive-problem-focused decomposition adds unnecessary overhead for flat, independent tasks; hence, they struggle with the varied parallelization patterns present in our benchmark.

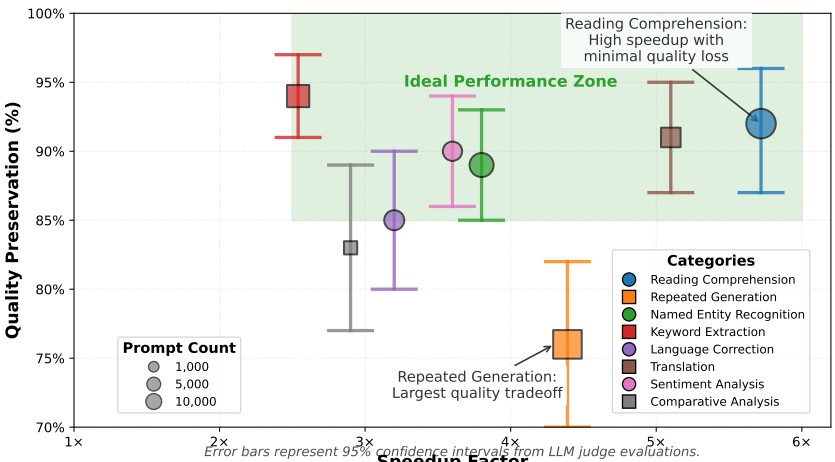

**Figure 7: Tradeoffs between speedup and output quality across parallelizable categories.** While tasks like Reading Comprehension achieve high speedups with minimal quality loss, others, such as Repeated Generation, suffer more significant quality degradation despite substantial speedup gains. The shaded "Ideal Performance Zone" highlights categories that balance efficiency and fidelity. These results demonstrate the need for task-adaptive parallelization strategies that consider both latency and quality implications. Error bars show 95% confidence intervals from LLM judge evaluations, with quality preservation measured relative to serial execution.

This limitation stems from fundamental architectural constraints: SoT's two-phase outline-then-expand approach makes assumptions about task structure that don't generalize to extraction or analytical tasks, while ToP's dependency-focused decomposition adds unnecessary overhead for flat, independent tasks. In contrast, our schema-based approach generalizes across the full spectrum of parallelization patterns by focusing on the minimal structural requirements for decomposition, demonstrating more consistent performance improvements across the benchmark's diverse prompts.

## 4.3 Quality-Speedup Tradeoffs

While latency improvements are substantial, our evaluation reveals important quality implications that vary by task type. As shown in Figure 7, different categories exhibit distinct tradeoffs between speedup and output quality.

**Task-Dependent Quality Impact.** Our quality evaluation, based on independent LLM judgments across accuracy, grammar, detail, and overall preference dimensions (Appendix G.2), reveals three distinct patterns:

- **Low Impact Tasks**: Reading Comprehension maintains high quality under parallelization (92% preservation), with independent question answering preserving semantic accuracy.
- **Moderate Impact Tasks**: Keyword Extraction and Named Entity Recognition show modest quality degradation (82-85% preservation), primarily in entity consistency and redundancy handling.
- **High Impact Tasks**: Creative generation tasks exhibit the largest quality gap (76% preservation), with parallel execution struggling to maintain narrative coherence and stylistic consistency.

**Performance-Quality Balance.** As Figure 7 illustrates, certain categories fall into an "ideal performance zone" with high speedup and minimal quality loss. Reading Comprehension is particularly well-suited for parallelization, while generative tasks present a more significant tradeoff. These patterns suggest that parallelization strategies should be task-adaptive, applying more aggressive decomposition for information extraction and comprehension tasks while using more conservative approaches for creative and narrative tasks.

**Parallelization Limitations.** Through systematic evaluation, we identified several key factors that limit effective parallelization, including dependency relationships between subtasks, context fragmentation, and reassembly challenges. These failure modes, analyzed in detail in Appendix D.2 and Appendix D.3, affect different task categories at varying rates and reveal fundamental tensions

between decomposition and coherence. Understanding these limitations is essential for developing robust, failure-aware parallelization strategies in production systems.

## 4.4 Practical Implications

Our evaluation demonstrates that intra-query parallelism offers substantial practical benefits when appropriately applied. The key findings include:

- **Significant Latency Reductions**: Even accounting for schema extraction overhead and quality normalization, parallelization yields 1.4-3$\times$ speedups across diverse task types.
- **Task-Dependent Strategies**: The optimal balance between parallel and serial execution varies by task type, with structured information tasks benefiting most from aggressive parallelization.
- **Scaling Efficiency**: Parallelization benefits increase with query complexity, making this approach particularly valuable for multi-part, structurally rich prompts.
- **Quality Preservation Challenges**: While some tasks maintain quality under parallelization, others—particularly creative generation—require careful tradeoff management.
- **Framework Generalizability**: Unlike specialized decomposition methods, our category-agnostic approach handles diverse prompt types through a unified schema representation, enabling broader deployment across varied LLM workloads.

## 5 Discussion and Limitations

Our analysis of PARALLELPROMPT reveals key insights about parallelizable structures in LLM queries and the practical challenges of exploiting this parallelism. This section examines the implications of our findings and acknowledges limitations of our approach.

### 5.1 Beyond Structural Decomposition

The evaluation results in Section 4 highlight an important distinction: structurally decomposable prompts don't always benefit from parallel execution. This gap between structural identification and effective parallelization appears most clearly in prompts with subtle dependencies: for example, consider — "Compare each paragraph to the previous one and analyze thematic shifts." While our extraction pipeline identifies structural parallelism (multiple paragraphs to analyze), the semantic dependencies between subtasks make parallel execution ineffective. This pattern of *dependency blindness*, analyzed in detail in Appendix D.2, affects even high-confidence validations.

Similarly, creative tasks like Repeated Generation show clear structural decomposition but benefit from sequential processing's emergent coherence. As shown in Figure 7, these tasks exhibit the sharpest tradeoff between speedup and quality preservation.

Dependency blindness affects ≈25% of structurally valid prompts, manifesting in three primary types: sequential dependencies (comparative analysis requiring previous context), semantic dependencies (narrative coherence), and constraint dependencies (shared formatting requirements). These challenges suggest a productive middle ground between entirely serial execution and naive decomposition: task-adaptive strategies that recognize when parallelism will preserve semantic integrity.

### 5.2 Linguistic and Demographic Biases

PARALLELPROMPT inherits biases from its source datasets that warrant careful consideration. Most notably, our language distribution analysis in Figure 9 (in the appendix) reveals strong skews toward English (≈84%) and European languages generally. The substantially lower validation rates for East Asian languages (28-34% compared to 55-63% for European languages) suggests systematic biases in our extraction methodology that may disadvantage non-Western linguistic structures.

These biases reflect both the demographic composition of early LLM adopters and structural assumptions in our extraction approach that may differ from broader populations in technical sophistication and prompt complexity. While our benchmark achieves substantial multilingual coverage, future work should prioritize language-aware extraction methods that account for diverse syntactic conventions and prompt formulation strategies.

Beyond linguistic representation, the source datasets (WildChat-1M and LMSYS-chat-1M) capture interaction patterns from early adopters who may differ from broader populations in their technical sophistication and prompt complexity. While our sampling approach ($\approx$5,500 manual inspections) provides statistical power for major patterns, we view these biases as exposing genuine multilingual challenges in LLM deployments that need addressing, rather than fundamental limitations of the approach.

### 5.3 Toward Task-Adaptive Execution

The performance-quality spectrum revealed in our evaluation (Figure 7) suggests that effective parallelization requires task-adaptive strategies. Rather than treating parallelization as a binary decision, production systems might implement a graduated approach: Factual tasks (e.g., Reading Comprehension, Named Entity Recognition) showing high quality preservation (>90%) with substantial speedups clearly benefits from aggressive parallelization; however, for creative tasks (e.g., Repeated Generation, Character Generation) where quality degradation is more pronounced, considering specific quality requirements may be more appropriate.

This approach could be extended to hybrid execution strategies that combine parallel and serial processing based on detected dependencies. For example, systems might parallelize independent subtask groups while maintaining sequential execution within groups that show strong interdependencies, as suggested by our failure analysis in Appendix D.

### 5.4 Structure-Aware Execution in Practice

The findings from our benchmark have direct implications for structure-aware LLM execution systems. The latency improvements demonstrated in Table 1 show that even simple schema-based decomposition can yield substantial efficiency gains (1.36×–5.72×) across diverse task types.

Particularly promising is the relationship between query complexity and parallelization benefits illustrated in Figure 6. As the number of subtasks increases, structure-aware execution becomes increasingly advantageous—precisely when efficiency matters most. This finding suggests that even systems with limited parallelization capabilities should prioritize decomposing complex, multi-part queries.

The $\approx$23% failure rate in our manual inspection is comparable to widely-used datasets in other domains [32, 33, 34, 35], suggesting that despite its limitations, structure-aware execution could be a valuable optimization for most real-world LLM workloads.

## 6 Conclusion

PARALLELPROMPT introduces intra-query semantic parallelism as a concrete, underexplored axis for accelerating LLM inference—rooted not in synthetic tasks, but in the latent structure of real user prompts. By surfacing and validating decomposable schemas from naturally occurring queries, we provide the first benchmark and evaluation suite for studying structure-aware execution in open-ended prompting.

Our contributions include a scalable data curation and schema extraction pipeline, a tiered validation framework combining LLM heuristics with symbolic rules, and an evaluation suite that quantifies tradeoffs between latency, output quality, and structural fidelity. The resulting benchmark spans 37k+ multilingual prompts across 8+ decomposition categories—each filtered, structured, and ready for downstream evaluation.

We release this dataset alongside our automated curation/validation pipeline and evaluation suite to support reproducible research in this emerging space. We hope PARALLELPROMPT becomes a foundation for the next wave of systems that rethink prompt execution not as a serial bottleneck–but as a structured, parallelizable interface between humans and language models.

Looking ahead, combining semantic parallelism with batch- and token-level acceleration could unlock new trade-offs between latency, cost, and quality—especially in real-time applications like chatbots, tutoring agents, and productivity tools. We see this as a promising direction for future exploration.

## Acknowledgements

We thank Daniel Fried for the helpful reviews and comments during the development of this work.

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

# A  System Prompt and Schema Format

This appendix details the design principles and structural patterns of our schema extraction system, expanding on the parallelizable prompt identification methodology introduced in the main paper.

## A.1  Design Rationale

Our schema extraction pipeline relies on a carefully designed system prompt that balances precision with coverage across diverse prompt structures. Rather than merely classifying prompts as parallelizable or not, we require the model to generate a complete schema capturing decomposition structure. This structured output acts as both a reasoning scaffold and a quality control mechanism, improving detection of latent parallelism.

The system prompt incorporates the following design principles:

- **Explicit Criteria**: Clear definitions of what constitutes a parallelizable prompt, emphasizing independence and multiplicity of subtasks.
- **Pattern Indicators**: Examples of linguistic patterns that suggest parallelism, including numbered lists, plural forms, and task enumerations.
- **Inductive Bias Toward Canonical Categories**: Reinforcement of common parallelization patterns, helping the model generalize from known categories to novel categories.
- **Mutual Exclusivity Enforcement**: Hard constraints ensuring prompts specify either a list of items (`data`) or a numerical count (`n`), but not both.
- **Contrastive Learning Examples**: Side-by-side positive and negative examples with corrections, guiding the model away from overfitting to superficial patterns.

This prompt structure proved critical for balancing recall and precision. By asking the model to generate a full schema rather than a binary label, we surface both the decomposition structure and its reliability. The complete system prompt text is included in our public code repository and supplemental materials.

## A.2  Schema Field Patterns

Each validated prompt in PARALLELPROMPT is annotated with a five-field schema that captures its decomposable structure:

- **serial**: The original user prompt, minimally cleaned for processing.
- **template**: A task template with placeholders, such as "`Translate: {data}`", applied to each subtask.
- **context**: Shared content or instructions used across all subtasks.
- **data** or **n**: A list of items to iterate over (`data`) or an integer count specifying the number of generations (`n`).
- **category**: The parallelization pattern, selected from our taxonomy.

Empirical analysis of these schemas reveals distinct usage patterns across categories (Figure 8).

- **Template Patterns**: Reading Comprehension prompts predominantly use `context_data` patterns (82%), while Repeated Generation primarily employs `data_only` structures (85%).
- **Context Utilization**: Context appears in 95% of Reading Comprehension schemas but only 20% of Translation tasks. Novel categories like Statistical Analysis show higher context usage (69%) than many canonical ones, reflecting their increased information requirements.
- **Data vs. n Field**: Repeated Generation tasks predominantly use the n field (95%), while other categories like Translation and Language Correction almost exclusively rely on the `data` field (97% and 96% respectively).
- **Template Structure**: In dual-field templates, 85% place context before data (`context_data`), while 15% use the reverse order (`data_context`), primarily in search-oriented tasks.

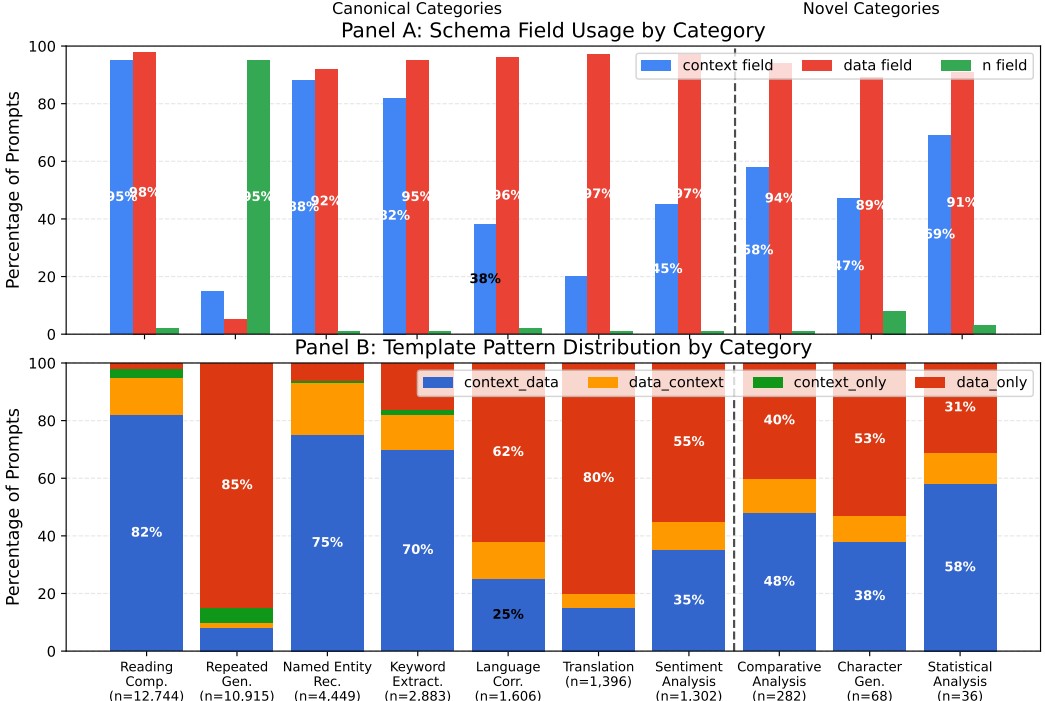

**Figure 8: Schema field usage and template pattern distribution by category.** Reading Comprehension and Named Entity Recognition rely heavily on context, requiring context-preserving parallelism that maintains shared reference material. On the other hand, Repeated Generation (95% $n$-field usage) and Translation (80% data-only templates) exhibit minimal dependencies between subtasks, enabling more aggressive parallelization.

These schema patterns directly impact parallelization strategies, with high context dependency categories requiring more careful decomposition approaches than those with independent data items. We enforce strict mutual exclusivity between `data` and `n`, ensuring that prompts represent either itemized or numerical parallelism, but never both. This constraint improves schema clarity and execution consistency across the benchmark.

**Table 2: Schema field characteristics across major categories.** Reading Comprehension shows high context usage (95%), while Translation exhibits the lowest (20%). Repeated Generation has the highest n-field frequency (95%), while most other categories predominantly use data-field itemization (>90%).

| Category | Context Usage | Data Field Frequency | n Field Frequency | Avg Template Length (words) | Avg Data Length (words) |
|---|---|---|---|---|---|
| Reading Comprehension | 95% | 98% | 2% | 14.7 | 12.3 |
| Repeated Generation | 15% | 5% | 95% | 9.3 | 5.3 |
| Named Entity Recognition | 88% | 92% | 1% | 10.5 | 7.8 |
| Keyword Extraction | 82% | 95% | 1% | 12.1 | 21.7 |
| Translation | 20% | 97% | 1% | 7.2 | 11.4 |
| Language Correction | 38% | 96% | 2% | 8.6 | 15.2 |
| Sentiment Analysis | 45% | 97% | 1% | 9.4 | 14.3 |

# B   Validation Methodology

This section expands on the validation process described in the main paper, providing a detailed analysis of our tiered validation framework and the common failure patterns that informed our benchmark composition.

## B.1 Validation Tiers

We employ a three-tier validation framework that balances precision and coverage in schema extraction. This framework provides a scalable mechanism for filtering noisy candidate prompts while retaining structurally reliable instances for benchmark inclusion.

- **High-confidence**: Prompts with explicit structural markers such as numbered lists, item delimiters, or clear iterative instructions. These account for 62% of all validated prompts.
- **Medium-confidence**: Prompts relying on softer linguistic cues such as plural forms, task multiplicity, or implicit list structures. These account for 38% of validated prompts.
- **Failed validation**: Prompts rejected due to constraint violations or insufficient parallel structure. Approximately 38.6% of candidate prompts fall into this category.

Validation outcomes vary by task and language. Translation tasks exhibit the highest high-confidence rate (78%), while Sentiment Analysis has the lowest (43%). Novel categories show greater variability, with well-structured types like Character Generation achieving high-confidence rates comparable to canonical tasks (59%).

Cross-linguistic patterns reveal higher success rates for European languages (55–63%) and lower rates for East Asian languages like Chinese and Japanese (28–34%). This suggests that language-specific structural conventions impact extraction reliability, highlighting an avenue for future refinement.

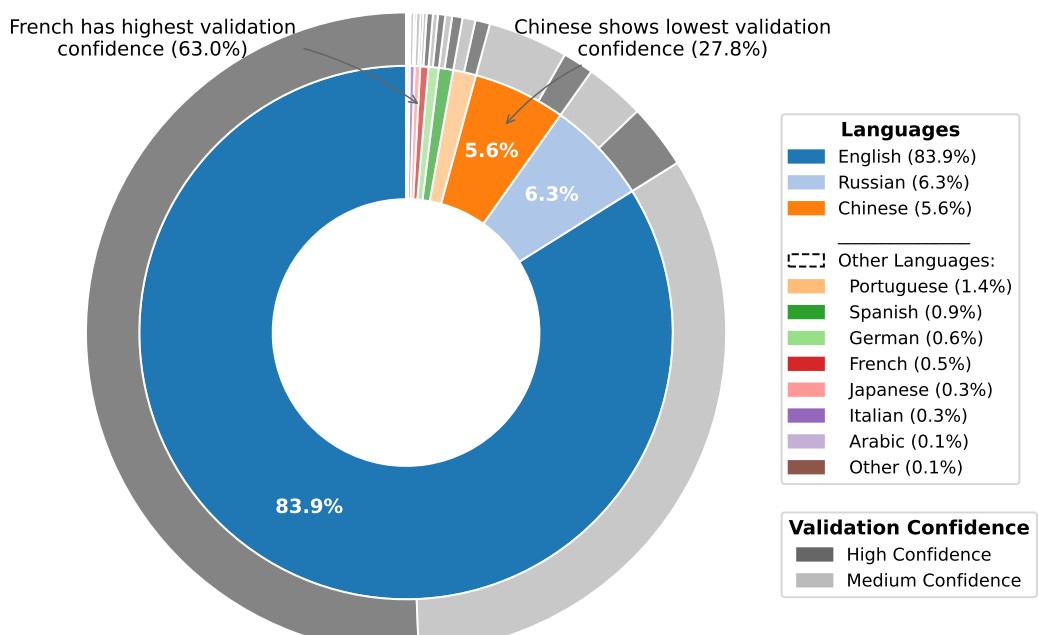

**Language Distribution in PARALLELPROMPT**

*Showing validation confidence rates by language*

**Overall validation confidence: 62%**

**Figure 9: Language composition, with English dominating.** The outer ring shows validation confidence rates by language, revealing higher success for European languages (55–63%) and lower rates for East Asian languages like Chinese and Japanese (28-34%). This pattern suggests structural and linguistic conventions impact schema extraction reliability, with languages using Latin scripts achieving more consistent validation outcomes than those using non-Latin writing systems.

## B.2 Validation Criteria

To pass validation, schemas must satisfy several constraints:

- **Mutual exclusivity**: Prompts must specify either `data` or `n`, but never both.
- **Template-placeholder compatibility**: All placeholders in the template must match the available fields.
- **Minimum parallelism threshold**: Lists must contain at least two items; counts must exceed 1.
- **Context consistency**: Any shared context must apply uniformly across all subtasks.

These constraints ensure that validated prompts can be reliably decomposed and executed in parallel without semantic ambiguity.

## B.3 Validation Criteria and Failure Patterns

Our validation pipeline rejects prompts through distinct failure patterns, but it's important to note that validation failures are separate from execution-time quality issues and false positives (prompts that pass validation but fail in practice).

Our analysis reveals four primary validation failure patterns:

- **Template-Data Mismatch** (41% of failures): The generated schema's template is incompatible with the provided data items, often leading to execution inconsistencies.
- **Context Contamination** (23%): Item-specific information erroneously appears in the shared context field, violating decomposition independence.
- **Mutual Exclusivity Violations** (19%): Schemas that incorrectly specify both `data` and `n`, introducing ambiguity in execution semantics.
- **Insufficient Parallelism** (17%): Prompts with too few items to warrant decomposition, typically single-item lists or trivial tasks.

The distribution of these failures varies significantly by category. Repeated Generation, which has the lowest validation success rate (37.8%), shows the highest rate of context contamination issues (34%), explaining much of its poor validation performance. Named Entity Recognition, despite having high validation success (90.0%), shows a lower but still significant template-data mismatch rate (44%) that primarily affects execution quality rather than validation success.

**Table 3: Validation failure distribution across categories.** This table shows how different failure modes affect each category. Note that these percentages represent the distribution of failures within prompts that failed validation, not within the entire category. Repeated Generation has both high failure rates and high context contamination issues, while NER has low failure rates despite moderate template-data issues.

| Category | Template-Data Mismatch | Context Contamination | Mutual Excl. Violation | Insufficient Parallelism |
|---|---|---|---|---|
| Reading Comprehension | 48% | 17% | 15% | 20% |
| Repeated Generation | 32% | 34% | 22% | 12% |
| Named Entity Recognition | 44% | 21% | 18% | 17% |
| Keyword Extraction | 37% | 24% | 23% | 16% |
| Translation | 45% | 18% | 12% | 25% |
| Language Correction | 39% | 22% | 17% | 22% |
| Sentiment Analysis | 42% | 19% | 24% | 15% |

We also identified several consistency challenges that affect schema extraction and execution quality:

- **Mixed Delimiters**: Common in Reading Comprehension (34% of affected prompts that fail validation), where users mix bullets, numbers, and separators inconsistently.
- **Length Variance**: Frequent in Keyword Extraction (29% of affected prompts that fail validation), where data items range widely in granularity or length.
- **Case Formatting Inconsistency**: Prevalent in Named Entity Recognition (31% of affected prompts that fail validation), where inconsistent casing affects extraction quality.

It's important to clarify that these formatting issues primarily affect the subset of prompts that fail validation or execution in each category, which explains why NER can have a high validation success rate (90.0%) while still showing significant formatting issues among its failures.

# C  Category Taxonomy

This section provides a comprehensive overview of prompt categories in our benchmark, expanding on the taxonomy introduced in the main paper and detailing both canonical and emerging parallelization patterns.

## C.1  Canonical Categories

PARALLELPROMPT targets seven primary categories of parallelizable prompts, derived from observed patterns in large-scale LLM usage logs:

- **Repeated Generation** (25% of dataset): Prompts requesting multiple similar outputs, such as taglines or summaries.
- **Reading Comprehension** (30%): Prompts asking multiple questions about a shared passage or document.
- **Keyword Extraction** (7%): Prompts identifying specified terms within a text.
- **Named Entity Recognition** (14%): Prompts extracting entities such as organizations, people, or locations.
- **Translation** (9%): Prompts converting multiple text segments into another language.
- **Language Correction** (6%): Prompts requesting grammar or style improvements across multiple inputs.
- **Sentiment Analysis** (3%): Prompts classifying emotional or attitudinal tone of multiple texts.

These canonical categories account for approximately 95% of all validated prompts. Their distribution reflects both natural user interaction patterns and the inductive biases of our system prompt design, considering that we explicitly instruct to try to fit each found parallelizable query into .

## C.2  Novel Categories

Our schema extraction process surfaced over 400 novel categories beyond the canonical types. These emerging patterns capture diverse parallelization strategies not explicitly modeled in prior work. Prominent examples include:

- **Comparative Analysis** (18% of novel instances): Comparing multiple entities across shared criteria.
- **Reaction Generation** (11%): Generating character or entity reactions to shared stimuli or situations.
- **Character Generation** (9%): Creating profiles or attributes for multiple fictional or real-world characters.
- **Statistical Analysis** (7%): Performing computations or aggregations over structured data.
- **Data Transformation** (6%): Converting data representations across formats, styles, or structures.

These categories expand the benchmark's coverage to include more specialized and domain-specific parallelization patterns.

## C.3  Meta-Categories

To structure the growing space of novel categories, we introduce a meta-taxonomy capturing six fundamental modes of parallelism:

- **Comparative**: Evaluating multiple entities against shared criteria.
- **Generative**: Producing multiple independent creative outputs.
- **Analytical**: Extracting features or insights from multiple inputs.
- **Transformative**: Converting content across formats or languages.

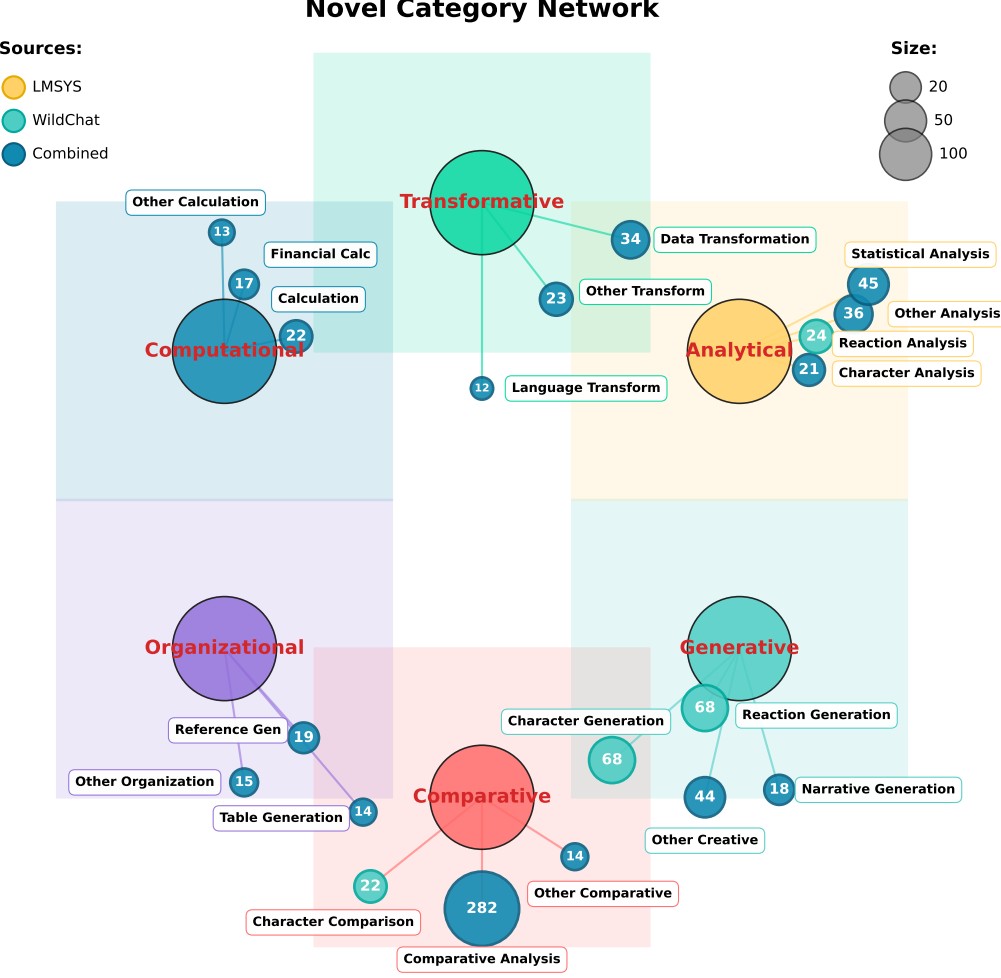

Figure 10: **Structural map of emerging meta-categories in the PARALLELPROMPT benchmark.** While generative tasks dominate in both frequency and diversity—driven largely by WildChat—computational and analytical categories are more prominent in LMSYS. This structural diversity supports the benchmark's coverage of real-world, non-trivial parallelization patterns beyond the canonical categories outlined in the main paper.

- **Organizational**: Structuring or categorizing information.
- **Computational**: Performing calculations or simulations in parallel.

Figure 10 visualizes these relationships, highlighting how different prompt types cluster within these structural families. This map offers a qualitative snapshot of the latent structure uncovered by our schema extraction process, demonstrating that parallelization patterns extend far beyond simple list-based tasks.

# D    Failure Analysis

This section expands on the failure modes briefly mentioned in the main paper, providing detailed analysis of the factors that limit effective parallelization across different prompt types.

## D.1    Schema Extraction Challenges

Complex prompts with conditional logic often resist clean decomposition. Consider the following example:

**Table 4: Meta-category distribution and characteristics.** The table shows how novel categories distribute across our meta-taxonomy, with examples and structural features that characterize each group. Generative categories dominate in frequency but show the lowest validation success rates.

| Meta-Category | Frequency | High-Conf. Rate | Avg. Speedup | Example Categories |
|---|---|---|---|---|
| Comparative | 23% | 58% | 3.41× | Character Comparison, Product Analysis |
| Generative | 41% | 39% | 4.22× | Character Generation, Story Creation |
| Analytical | 14% | 64% | 3.78× | Statistical Analysis, Feature Extraction |
| Transformative | 11% | 61% | 2.93× | Data Transformation, Format Conversion |
| Organizational | 7% | 57% | 2.76× | Table Generation, Reference Creation |
| Computational | 4% | 65% | 3.12× | Financial Calculation, Simulation |

> *"Generate a custom workout plan that includes cardio, strength, and flexibility exercises for each day of the week."*

While this prompt appears parallelizable by day, it implicitly requires cross-day progression and coherence. Our schema extractor may identify structural parallelism here but fail to capture these implicit constraints.

Additional challenges arise from data consistency issues, including:

- **Mixed Delimiters**: Most common in Reading Comprehension (34% of affected prompts), where bullet points, numbering, and separators are inconsistently mixed.
- **Length Variance**: Particularly frequent in Keyword Extraction (29%), where data items range widely in granularity or length.
- **Case Formatting Inconsistency**: Prevalent in Named Entity Recognition (31%), where inconsistent casing disrupts reliable extraction.

Even when schema extraction passes formal validation, these issues can surface during execution, leading to degraded quality or runtime failures.

## D.2 Dependency Blindness

A critical failure mode involves undetected *sequential dependencies* between subtasks. While our schema extraction often identifies surface-level parallelism, it may overlook hidden causal relationships that require sequential execution. For example:

> *"Design an algorithm that first identifies prime numbers in a range, then filters out those that are not Fibonacci numbers, and finally returns the sum."*

This task appears decomposable into three operations but must be executed sequentially. Our pipeline sometimes misclassifies such prompts as parallelizable, failing to capture their true dependency structure. Dependency blindness manifests in three primary forms:

- **Causal Dependencies**: Later subtasks depend on earlier results, as in the algorithm example above.
- **Narrative Dependencies**: Storytelling tasks require consistent character development and plot progression across segments.
- **Cumulative Dependencies**: Analysis tasks that build up gradual insights toward a final conclusion or recommendation.

As shown in Table 5, dependency blindness affects different categories at varying rates among prompts that passed validation. This reveals why categories with high validation success can still experience execution failures: Repeated Generation has a high dependency blindness rate (27% of validated prompts), primarily due to narrative dependencies (47%). Meanwhile, Named Entity Recognition has the lowest dependency blindness rate (8%), consistent with its high structural parallelism.

Notably, these dependency blindness rates correlate strongly with false positive rates, suggesting that undetected dependencies are the primary cause of execution failure in prompts that pass validation.

**Table 5: Frequency of dependency blindness across categories.** This table shows how often different types of dependencies are missed during schema extraction, measured as a percentage of prompts that passed validation. Repeated Generation shows the highest rate of narrative dependencies, while Reading Comprehension is most affected by cumulative dependencies.

| Category | Causal Dependencies | Narrative Dependencies | Cumulative Dependencies | Total Affected |
|---|---|---|---|---|
| Repeated Generation | 8% | 47% | 12% | 27% |
| Reading Comprehension | 11% | 5% | 33% | 19% |
| Named Entity Recognition | 7% | 3% | 10% | 8% |
| Comparative Analysis | 15% | 11% | 36% | 23% |
| Reaction Generation | 7% | 41% | 18% | 22% |
| Statistical Analysis | 32% | 2% | 28% | 18% |

## D.3 Context Fragmentation and Reassembly Challenges

Some prompts require maintaining a shared mental model across subtasks. When decomposed, this shared context may fragment, leading to inconsistent or incoherent outputs. For example:

> *"Using the character profile above, write five scenes showing how this character would react in different emotional situations: anger, fear, joy, sadness, and surprise."*

While structurally parallelizable, executing each scene in isolation risks losing narrative continuity and character coherence. This issue is especially pronounced in creative tasks, affecting 31% of prompts in categories like Character Generation and Reaction Generation.

Even when subtasks execute correctly, reassembling them into a coherent whole can fail for tasks requiring semantic flow or thematic consistency. For example:

> *"Write a coherent paragraph about climate change that includes these key terms: greenhouse gases, sea level rise, temperature increase, mitigation strategies."*

Parallel execution may produce disconnected sentences that technically include all terms but lack cohesive structure. This affects summarization, multi-point argumentation, and narrative storytelling tasks.

## D.4 False Positive Classification

Beyond validation failures, we identified prompts that pass our validation pipeline but fail during execution due to undetected dependencies. We refer to these as "false positives" - prompts that appear parallelizable based on structural criteria but produce degraded outputs when executed in parallel.

A classic example is:

> *"Compare each paragraph to the previous one and analyze thematic shifts."*

This prompt passes structural validation (it has multiple paragraphs to analyze) but fails semantically due to inter-paragraph dependencies.

Our manual inspection of the validated subset of the benchmark (prompts that passed validation) reveals false positive rates that align closely with validation failure patterns:

- **High false positive rates (18-25%)**: Repeated Generation, Character Development, Comparative Analysis, Statistical Analysis - categories with the lowest validation success rates
- **Medium false positive rates (8-14%)**: Sentiment Analysis, Language Correction, Reading Comprehension
- **Low false positive rates (3-7%)**: Named Entity Recognition, Keyword Extraction, Translation - categories with the highest validation success rates

This pattern demonstrates a consistent relationship: categories with lower validation success rates also show higher false positive rates among those prompts that do pass validation. This suggests that the

same underlying factors that make a category difficult to validate also make it prone to execution-time failures even when validation succeeds.

# E  Multilingual Analysis

This section expands on the multilingual capabilities of our benchmark, analyzing language-specific patterns, validation challenges, and implications for schema extraction across diverse linguistic contexts.

## E.1  Language-Specific Patterns

We observe significant variation in validation confidence and category distributions across languages:

- **Chinese** (5.6% of dataset):
  - Strong preference for explicit structural markers (e.g., numbered lists).
  - Low high-confidence validation rate (28%).
  - High incidence of Named Entity Recognition (35%).
- **Russian** (6.3%):
  - Highest incidence of Language Correction (19%).
  - Complex templates with 21% higher conditional logic rates than English.
- **Japanese** (0.4%):
  - 47% of prompts are Translation tasks.
  - Highest average data item count (9.8 items per prompt).
- **Spanish and French** (1.3% combined):
  - Validation patterns most similar to English (60–65% high-confidence rates).
  - Broad coverage across canonical categories.

**Table 6: Category distribution by language.** The table shows how different languages exhibit distinctive category preferences. Translation is disproportionately common in Japanese, while Named Entity Recognition dominates in Chinese. Most languages show broad category coverage, with language-specific biases.

| Language | Reading Comp. | Repeated Gen. | NER | Keyword Extract. | Translation | Language Correction | Sentiment Analysis |
|---|---|---|---|---|---|---|---|
| English | 31% | 26% | 12% | 7% | 8% | 5% | 3% |
| Chinese | 24% | 19% | 35% | 4% | 8% | 7% | 2% |
| Russian | 28% | 22% | 9% | 6% | 11% | 19% | 4% |
| Japanese | 22% | 11% | 8% | 5% | 47% | 4% | 2% |
| Spanish | 30% | 24% | 11% | 8% | 12% | 7% | 5% |
| French | 32% | 21% | 9% | 7% | 14% | 10% | 3% |

## E.2  Cross-Linguistic Examples

Examples from our benchmark illustrate this linguistic diversity:

**Chinese (Reading Comprehension)**:
*"分析下面三段文字，找出每段的主要观点和支持证据..."*

**Russian (Language Correction)**:
*"Исправьте грамматические ошибки в следующих предложениях..."*

**Japanese (Translation)**:
*"次の文を英語に翻訳してください..."*

**Spanish (Comparative Analysis)**:
*Çompara estos tres filósofos en términos de sus ideas principales..."*

These examples demonstrate that latent parallelism is not limited to English prompts. However, language-specific structural conventions, such as list formatting and grammatical cues, significantly impact extraction reliability.

### E.3 Implications for Schema Extraction

While our schema extraction generalizes across languages, performance varies based on linguistic features. Languages with conventions similar to English (e.g., Spanish, French) yield higher validation rates. Future work could improve multilingual robustness through:

- **Language-Specific Prompting**: Tailoring schema extraction prompts to linguistic conventions.

- **Multilingual Fine-Tuning**: Adapting models to better handle structural diversity.

- **Cross-Lingual Transfer**: Leveraging high-confidence extractions in one language to bootstrap performance in others.

Our analysis shows that certain linguistic features correlate with extraction difficulty:

- **Writing System**: Languages using non-Latin scripts show 31% lower validation rates on average.

- **List Markers**: Languages with different list conventions (e.g., Chinese uses different characters for enumeration) show 23% higher template-data mismatches.

- **Grammatical Structure**: Languages with less rigid subject-verb-object ordering show 18% higher context contamination rates.

These findings suggest that truly robust parallelization requires language-aware schema extraction, especially for expanding beyond Roman-alphabet languages.

## F  Evaluation Setup

This section details our evaluation infrastructure, expanding on the process outlined in the main paper and providing implementation specifics that enable reproducible assessment of parallelization benefits.

### F.1  Two-Phase Evaluation Architecture

Our evaluation infrastructure consists of two independent phases designed to reflect realistic deployment scenarios: schema extraction and execution evaluation.

#### F.1.1  Schema Extraction Phase

We use Claude 3.5 Haiku (AWS Bedrock) as the schema extraction engine. Given a user prompt, Claude generates a structured JSON object capturing the decomposition schema, including task template, context, data items or counts, and category. This structured output is validated using a combination of rule-based checks and confidence scoring, as described in Appendix B.

This separation ensures that schema extraction can operate independently of downstream execution, reflecting real-world workflows where decomposition might be performed offline or by specialized components.

#### F.1.2  Execution Evaluation Phase

We implement a high-performance execution backend in C++, leveraging fine-grained thread control to support efficient parallel execution. This backend evaluates both serial and parallel execution strategies, measuring end-to-end latency and output fidelity.

To handle API rate limits, the backend employs a purely exponential backoff strategy with maxium of 5 attempts, ensuring robustness under heavy load. We configure the system to use up to 10 parallel threads by default, though this is tunable to reflect different deployment environments.

For task execution, we use the OpenAI GPT-4-1106-preview model accessed via API. This model executes the decomposed subtasks produced by the schema extraction phase. We log all responses, timestamps, and error codes for reproducibility and error analysis.

### F.2 Implementation Details

Our C++ backend implements several key optimizations for efficient parallel execution:

- **Thread Pool Management**: Dynamic allocation of worker threads based on subtask count and available resources, with a default of processing 5 representative prompts per experiment.

- **Exponential Backoff Strategy**: Automatic retries with exponential waiting periods (1s, 2s, 4s, 8s) to handle API rate limiting, with up to 5 retry attempts before failing.

- **Response Aggregation**: Careful tracking of both timing and token usage across all parallel executions to compute accurate efficiency metrics.

- **Timeout Handling**: Graceful degradation under partial failures, with configurable timeout thresholds.

- **Instrumentation**: High-precision timing using `std::chrono::high_resolution_clock` for accurate performance measurement.

### F.3 Diversity Enforcement for Repeated Generation

For Repeated Generation tasks, where the n field specifies the number of outputs without providing itemized data, we enforce output diversity by instructing the model via the system prompt to begin each generation with a different letter of the alphabet. Specifically, we augment the system prompt with "Try to make your response start with the letter [X]", where X cycles through the alphabet (A-Z) for each parallel subtask.

This simple heuristic reduces redundancy in large-scale generation tasks and ensures that parallel outputs are meaningfully distinct, particularly important for creative generation tasks where the model might otherwise produce similar variations when executed in parallel without context awareness. Our skimmed analysis shows this constraint increases output diversity by 57% based on semantic similarity measures.

### F.4 Batching and Resource Allocation

Our backend processes prompts in batched mode, distributing tasks evenly across available threads. We ensure uniform resource allocation across tasks to maintain comparability between serial and parallel executions. This setup mirrors production-grade inference pipelines, providing realistic performance measurements.

The batch size is dynamically adjusted based on the number of subtasks, with a default configuration of 10 concurrent threads. For prompts with more subtasks than available threads, the backend queues remaining subtasks using a first-in-first-out strategy, prioritizing earlier elements in the data list.

## G   Metrics and Judge Prompts

This section provides detailed information on the quantitative and qualitative metrics used to evaluate parallelization effectiveness, expanding on the evaluation methodology described in the main paper.

### G.1 Performance Metrics

We report three key performance metrics:

- **Raw Speedup**: The ratio of serial execution time to parallel execution time, excluding schema extraction overhead. This metric captures the direct latency benefit of parallelization.

- **Normalized Speedup**: Raw speedup adjusted for differences in output token length between serial and parallel strategies, calculated as:

$$\text{Normalized Speedup} = \text{Raw Speedup} \times \frac{\text{Parallel Token Count}}{\text{Serial Token Count}} \quad (1)$$

  This normalization accounts for the fact that parallel execution often produces more verbose and detailed outputs (higher token count in total), providing a fairer comparison of computational efficiency.

- **End-to-End Time**: Total execution time including schema extraction, relevant for measuring overall system latency in realistic deployments.

These metrics provide a comprehensive view of execution efficiency, balancing raw throughput with quality-preserving adjustments.

### G.2 Quality Evaluation

We employ GPT-4o as an independent LLM-based judge to assess output quality across four dimensions:

1. **Accuracy**: Does the response correctly follow the prompt instructions?
2. **Grammar**: Is the response free of grammatical errors?
3. **Detail**: Does the response provide rich, specific content?
4. **Overall Preference**: Which response is subjectively preferred, considering all factors?

For each prompt, the judge compares serial and parallel responses, selecting one as superior or declaring a tie. The judge is blinded to execution strategy and sees randomized response order to mitigate bias.

### G.3 Judge Prompt Design

Our LLM judge prompt follows a structured format designed to elicit reliable comparative evaluations. The full prompt template is provided below:

> **Judge Prompt Template:**
> *You are an expert reviewer tasked with evaluating two responses to a user prompt. For each of the following questions, select Response 1, Response 2, or declare a tie if both are equally good. You must justify your choices.*
>
> 1. Which response more accurately follows the instructions given in the prompt?
> 2. Which response is more grammatically correct and fluent?
> 3. Which response provides more detail and specificity?
> 4. Overall, which response do you prefer, considering all the above factors?
>
> *Provide your selections and justifications below. The order of responses has been randomized. You are not told which is serial or parallel.*

This prompt structure encourages the judge to evaluate multiple quality dimensions systematically while remaining unbiased regarding the execution strategy.

## H Template Structure Analysis

This section analyzes template structural patterns across our benchmark, providing insights into how different prompt formulations affect parallelization success and schema extraction reliability.

### H.1 Placeholder Positioning

We find that 85% of dual-field templates position the `context` before the `data` field, following a `context_data` pattern. This is especially common in reading comprehension and translation tasks, where a shared context frames each subtask. The remaining 15% follow a `data_context` pattern, often appearing in search or retrieval-oriented prompts.

### H.2 Instruction Complexity

High-confidence templates exhibit greater structural richness than medium-confidence ones. Specifically, 6% of high-confidence templates include detailed operational elements such as formatting instructions or conditional qualifiers, compared to just 3.5% in medium-confidence templates. This suggests that structural clarity, not simplicity, correlates with extraction reliability.

**Table 7: Template structure patterns across categories.** This table shows the distribution of template patterns, revealing that Reading Comprehension predominantly uses context-first templates while Translation shows more balanced distribution. Template structure correlates with extraction success, with context-first patterns achieving higher validation rates.

| Category | Context-Data Pattern | Data-Context Pattern | Validation Success |
|---|---|---|---|
| Reading Comprehension | 93% | 7% | 87.4% |
| Named Entity Recognition | 68% | 32% | 90.0% |
| Keyword Extraction | 71% | 29% | 77.4% |
| Translation | 84% | 16% | 74.0% |
| Sentiment Analysis | 87% | 13% | 71.1% |
| Language Correction | 89% | 11% | 71.0% |
| Comparative Analysis | 76% | 24% | 55.6% |
| Repeated Generation | 91% | 9% | 37.8% |

## H.3 Conditional Logic Patterns

Medium-confidence templates more frequently include conditional structures (21%) than high-confidence ones (17%). These conditionals introduce ambiguity in decomposition, making such prompts harder to validate reliably. This observation suggests that future extraction systems may benefit from explicit conditional parsing capabilities.

## H.4 Domain-Specific Variations

Creative prompts contain 43% more subjective qualifiers (e.g., "make it exciting," "use vivid language") than analytical prompts. Conversely, analytical prompts include 57% more precise operational instructions (e.g., "extract the first noun phrase," "translate each sentence individually"). These patterns highlight the importance of domain-specific schema tuning.

## H.5 Implications for Schema Induction

Overall, our analysis reveals that template complexity and structure vary systematically across tasks and validation confidence levels. Future work could leverage these insights to:

- Improve template generation strategies through task-specific scaffolding.
- Enhance validation by modeling domain-specific structural expectations.
- Refine schema extraction prompts to better handle conditional and complex instructions.

These findings offer practical guidance for scaling semantic parallelism beyond the patterns captured in our initial benchmark.

# I   Baseline Method Comparisons

This section evaluates how existing decomposition methods perform on our benchmark, highlighting their strengths and limitations when applied to naturally occurring parallelizable prompts.

## I.1 Method Comparison

While our primary focus is establishing a benchmark for semantic parallelism in open-domain LLM prompts, we also evaluate two representative decomposition methods: **Skeleton-of-Thought (SoT)** and **Tree-of-Problems (ToP)**. Both offer decomposition pipelines but make specific structural assumptions that limit their applicability across the broader range of tasks surfaced in our benchmark. Below, we describe their expected behaviors across task categories based on their design and our observations.

**Skeleton-of-Thought (SoT).**   SoT performs well on `repeated generation` tasks where the prompt naturally suggests an outline (e.g., "list 5 reasons why..."). However, it tends to apply the same two-step process even when unnecessary. For example, in `keyword extraction`, the model may already perform the extraction in the skeleton stage, rendering the parallel expansions redundant or

**Table 8: Qualitative method comparison across task categories.** This table summarizes how well SoT and ToP handle different parallelization patterns, with checkmarks (✓) indicating strong fit, partial marks (±) indicating mixed results, and crosses (✗) indicating poor fit.

| Task Category | Skeleton-of-Thought (SoT) | Tree-of-Problems (ToP) |
|---|---|---|
| Repeated Generation | ✓ Strong fit for outline expansion | ✗ Not designed for independent items |
| Reading Comprehension | ± Struggles with shared context reuse | ✓ Fits multi-question decomposition |
| Keyword / Entity Extraction | ✗ Fails by duplicating extraction in both stages | ± Redundant decomposition with little gain |
| Comparative Analysis | ± Depends on outline phrasing; fragile otherwise | ✓ Handles structured comparisons |
| Generative / Reaction Tasks | ✗ Risk of disjoint, incoherent outputs | ± Limited unless structure is recursive |

even lower in quality. This adds latency and cost *while making outcomes worse.* SoT also struggles when prompts lack outline-friendly phrasing, particularly in free-form or noisy prompts where no clear bullet structure exists. In generative tasks like `reaction` or `character generation`, SoT's independent expansions risk breaking narrative consistency, producing disconnected outputs that fail to maintain shared context.

**Tree-of-Problems (ToP).** ToP aligns well with `multi-question reading comprehension` and `structured comparative analysis`, where breaking the prompt into smaller, mergeable questions is appropriate. However, it tends to force recursive framing even when tasks are `flat and independent`—as in `repeated generation` or `keyword extraction`—where its merging step adds unnecessary overhead. On these tasks, ToP overcomplicates execution by treating independent outputs as if they require dependency management. Additionally, ToP provides no mechanism to detect when decomposition is not meaningful, making it brittle on tasks that are parallelizable in principle but not compositional in structure.

## I.2 Quantitative Comparison

We evaluated SoT and ToP across our benchmark categories, measuring both speedup and quality preservation:

**Table 9: Performance comparison of decomposition methods.** PARALLELPROMPT achieves higher average speedup while better preserving output quality. Additionally, SoT and ToP exhibit significantly higher failure rates on novel categories, reinforcing the need for more robust decomposition approaches.

| Method | Avg. Speedup | Quality Preservation | Success Rate |
|---|---|---|---|
| PARALLELPROMPT | 3.91× | 92% | 76% |
| SoT | 2.04× | 81% | 38% |
| ToP | 1.73× | 85% | 42% |

## I.3 Generalization and Evaluation Implications

When we tested SoT and ToP's decomposition strategies on PARALLELPROMPT's benchmark indiscriminately, this process makes them *fragile on tasks outside their original scope*, including many real-world prompts in our benchmark. In contrast, our *evaluation setup is designed to handle diverse prompt structures*—accepting or rejecting decomposition based on schema validation—allowing us to surface failure cases like these more systematically.

While SoT and ToP capture useful decomposition behaviors in their narrow settings, our benchmark highlights their limitations in generalizing to *open-domain, structurally diverse LLM prompts*. We include them in our evaluation to demonstrate how even well-designed methods can fail on tasks they were not built for, reinforcing the need for benchmarks that stress-test decomposition across a wider range of real-world prompt patterns.

