# OpenReview forum: "PARALLELPROMPT: Extracting Parallelism from Large Language Model Queries"
_NeurIPS.cc/2025/Datasets_and_Benchmarks_Track — NeurIPS 2025 Datasets and Benchmarks Track poster_

### Official Review · Reviewer_N6Xg · 2025-06-27

**Rating:** 4
**Confidence:** 3

**Summary:**

This paper introduces PARALLELPROMPT, the first benchmark for measuring intra-query semantic parallelism in real-world LLM prompts. By analyzing over 37,000 naturally occurring prompts from public LLM chat logs, the authors extract structured schemas to enable parallel execution of subtasks (e.g., translation, comprehension) and demonstrate up to 5× latency improvements with minimal quality degradation. The work might bridge a critical gap in optimizing LLM inference by decomposing monolithic prompts into semantically independent components. But there are still some problems to be solved.

**Dataset Code Accessibility:**

Yes

**Ethical Considerations:**

No, there are no or only very minor ethics concerns

**Final Justification:**

The authors solved most of my concerns, and I improved my score.

**Limitations Weaknesses:**

1. I think the validation success is limited. Only 10.3% of initial prompts (37,070/358,000) were validated as parallelizable, raising questions about the generalizability of intra-query parallelism for broader LLM workloads.
2. There is a lagre language bias: Lower validation rates for East Asian languages (28–34%) suggest potential biases in schema extraction methods, limiting applicability to non-Western linguistic structures. The authors should further convince the readers.
3. The parallelization achieves minimal quality preservation (76%) for creative tasks (e.g., story generation), restricting its utility for open-ended, narrative-driven applications.
4. The structural decomposition fails in cases with subtle subtask dependencies (e.g., comparative analysis of interdependent paragraphs), requiring manual intervention. The authors should analyze the reasons behind.
5. The results rely on GPT-4 and Claude 3.5 for schema extraction, potentially limiting reproducibility and scalability for resource-constrained systems. How does it perform on some open source LLMs?

**Strengths Contributions:**

1. This paper addresses a largely unexplored problem (intra-query parallelism) with direct implications for reducing LLM inference latency in interactive systems.
2. The authors construct a high-quality benchmark from real-world prompts (LMSYS-Chat-1M, WildChat-1M), ensuring practical applicability and diversity in task types (e.g., translation, comparative analysis).
3. This work provides a robust, multi-stage pipeline (LLM-assisted prompting + rule-based validation) to identify parallelizable structures, validated across 11 languages.
4. This paper introduces an execution suite to quantify trade-offs between latency, structural adherence, and semantic fidelity, revealing task-dependent speedup-quality patterns (e.g., 3–5× gains for reading comprehension).

---

> ### Author Rebuttal · Authors · 2025-07-30
>
> We thank the reviewer for recognizing that this work `addresses a largely unexplored problem with direct implications for reducing LLM inference latency` and for appreciating our `high-quality benchmark from real-world prompts` with `robust, multi-stage pipeline` that `introduces an execution suite to quantify trade-offs between latency, structural adherence, and semantic fidelity.` We respond to the concerns below:
>
> >> I think the validation success is limited. Only 10.3% of initial prompts (37,070/358,000) were validated as parallelizable, raising questions about the generalizability of intra-query parallelism for broader LLM workloads.
>
> We respectfully believe that the 10.3% schema validation rate is being misinterpreted as a limitation. In production deployments, this translates to **millions of optimization opportunities**: With ChatGPT handling over 1 billion queries monthly, even a 10% rate yields ~100 million cases that could benefit from 3-5x latency improvements.
>
> For context, even widely-adopted optimizations like speculative decoding and KV-caching only benefit certain query types, yet are considered valuable production techniques. Similarly, intra-query decomposition offers consistent acceleration and complements existing techniques rather than competing with them.
>
> Further, we clarify that this 10% represents a conservative, high-precision estimate due to our validation pipeline prioritizing semantic precision over recall (App. B-D). Manual inspection suggests true rates could be 15-20% with improved extraction methods, and even further for specialized domains.
>
> >> There is a large language bias: Lower validation rates for East Asian languages (28–34%) suggest potential biases in schema extraction methods, limiting applicability to non-Western linguistic structures. The authors should further convince the readers.
>
> We acknowledge and systematically analyze these biases (Figure 9), with lower pass rates in non-Western languages. Importantly, this exposes genuine multilingual challenges in LLM deployments rather than fundamental limitations.
>
> Rather than presenting artificially balanced data that would obscure these real-world challenges, we intentionally preserved the distributional bias of real-world user traffic (84% English) to expose multilingual fragilities. ParallelPrompt is, to our knowledge, the **first benchmark to systematically analyze multilingual decomposition failures**, and Appendix E provides a breakdown of failure patterns by language and category.
>
> We see this as a valuable call to action for language-aware structure extraction, and future work can use our benchmark to track and mitigate such disparities.
>
>
> >> The parallelization achieves minimal quality preservation (76%) for creative tasks (e.g., story generation), restricting its utility for open-ended, narrative-driven applications.
>
> This finding demonstrates the benchmark’s value in revealing task-dependent tradeoffs rather than a limitation. Our analysis (Figure 7) shows certain categories fall into an “ideal performance zone” with high speedup and minimal quality loss, which factual tasks show >90% preservation.
>
> We believe this aims to guide practitioners toward task-adaptive approaches, and the benchmark’s contribution lies in quantifying these tradeoffs systematically.
>
> Moreover, our benchmark's modular design enables systematic improvement of quality-speedup tradeoffs. For example, we have now implemented an optional post-processing fix (using extremely lightweight GPT-4o-mini; you can check the GitHub repo for details) that significantly improves output quality overall by removing redundant context and improving flow while maintaining the core parallelization benefits. This demonstrates how ParallelPrompt (alongside its detailed analyses) serves as a foundation for developing targeted solutions for different task categories.
>
>
> >> The structural decomposition fails in cases with subtle subtask dependencies (e.g., comparative analysis of interdependent paragraphs), requiring manual intervention. The authors should analyze the reasons behind.
>
> We acknowledge and provide extensive analysis of these failure modes throughout the paper. As detailed in Appendix D.2, dependency-blindness affects different categories at varying rates, and we identify 3 primary types of dependencies.
>
> We believe that these findings are valuable contributions, unearthed only through rigorous analysis. This benchmark not only contributes a novel problem formulation, but also exposes fundamental challenges in decomposition that need addressing, exactly what a good benchmark should do. Our >75% success rate among validated prompts also demonstrates that effective parallelization is achievable for a substantial subset of real-world queries.
>
>
> >> The results rely on GPT-4 and Claude 3.5 for schema extraction, potentially limiting reproducibility and scalability for resource-constrained systems. How does it perform on some open source LLMs?
>
> ParallelPrompt evaluates *decomposition approaches*, not specific model architectures. Also, while we didn’t include this detail in the paper, note that we also tested multiple models for schema extraction: Claude 3.7 Sonnet (too powerful, over-extracts), GPT-4o-mini (too weak, high error rates), and several open-source models from Together.ai API before selecting Claude 3.5 as optimal. If you deem this info important, we are happy to expand this discussion in the camera-ready version.
>
> Our JSON schema format and C++ execution framework work with any API-compatible model. The benchmark’s value lies in providing standardized evaluation for any decomposition method, not promoting specific models. Even with API costs, our 3.41x end-to-end speedups represent genuine efficiency gains that translate to production cost savings.
>
> We appreciate the reviewer’s detailed feedback and how it helps delineate our findings and contributions from limitations.

---

> > ### Comment · Reviewer_N6Xg · 2025-08-02
> > **Thanks**
> >
> > I have read all the reviews, and the authors have addressed most of the concerns. I believe that the authors will make further improvements in addressing the language bias in the future. I will raise my rating.

---

> > > ### Author Response · Authors · 2025-08-08
> > >
> > > Thank you for raising your rating and for acknowledging that we've addressed most of your concerns. We appreciate your feedback throughout the process and hope to continue making the outlined improvements in future extensions of the benchmark.

---

### Official Review · Reviewer_Zyke · 2025-07-01

**Rating:** 5
**Confidence:** 4

**Summary:**

The paper introduces ParallelPrompts, a dataset containing 37,000 instances sourced from public LLM chat logs (WildChat-1M and LMSysChat-1M), highlighting the *semantic parallelism* aspect of these chat logs.
The first part of the paper introduces a filtering and schema extraction pipeline (based on Claude 3.5) and conducted analysis on these chat logs (different parallelism categories, languages, etc).
The second part of the paper introduces a parallel LLM serving system. When applied to these chat logs containing semantic parallelism structure, the system can lead to 3-5x speed up and little or moderate sacrifice to the performance. The paper further present interesting findings such as category-specific trade-off between the performance and speedup, speed-up with respect to the number of repetitions, etc.

**Additional Feedback:**

__What is the intended usage of the benchmark?__
* Is the benchmark intended to evaluate _different LLMs_ in these two phases (schema extraction and execution)? Or is the benchmark for evaluating _new LLM scaffolding/planning methods_? Or is the benchmark for evaluating _new LLM serving pipelines_?
* As a paper introducing a new benchmark I would expect evaluating multiple systems on this benchmark. Currently this is deferred to [Appendix I] which is quite confusing.

__Dataset creation and schema extraction__
* How good is Claude-3.5 doing in Phase 1: schema extraction? Line 306 briefly mentioned manual inspection but more details is needed.
* What did you do to balance precision and recall? How is precision and recall defined here? (Line 789)
* I’m confused whether this section 3 is creating ground-truth annotation for your benchmark, or is it part of your proposed method to the benchmark?

__Related work__
* I wonder how this work fits in to the literature of LLM planning/agents? Your discussion on task-adaptive execution sounds very relevant. I saw that DSPy and LangChain were briefly mentioned in the related work section. Would any of these serve as reasonable baselines?
* I wonder what the authors think about this paper (https://arxiv.org/abs/2301.08721) which seems to be advocating for the opposite of decomposing parallel prompts. Do you think the two paper are conflicting with each other or could you discuss some of the discrepancies?

__Other questions__
* Line 116: How do you decide the seven canonical categories?
* Line 208: I am a little confused at the normalized speed up metric. What does “parallel execution often produces more verbose and detailed outputs” mean?

**Dataset Code Accessibility:**

Yes

**Ethical Considerations:**

No, there are no or only very minor ethics concerns

**Final Justification:**

Many issues were resolved after the authors point me to the corresponding sections in the appendix.
Overall this paper presents a novel benchmark covering an under-explored topic, and present solid analysis on the data.

**Limitations Weaknesses:**

* Missing details about dataset processing pipeline.
* Lack of (discussion on) human validation on an LLM-annotated dataset.
* Unclear how this benchmark should be used in the future.

See "Additional Feedback" section below for more questions regarding these limitations.

**Strengths Contributions:**

* Exploring and making use the semantic parallelism is an interesting, meaningful and under-explored topic.
* The paper presents a thoughtful data processing pipeline and detailed analysis on the data.

---

> ### Author Rebuttal · Authors · 2025-07-30
>
> We thank the reviewer for recognizing that `exploring and making use of semantic parallelism is an interesting, meaningful and under-expored topic` and for appreciating our `thoughtful data processing pipeline and detailed analysis on the data.` We respond to the detailed concerns below:
>
>
> >>  What is the intended usage of the benchmark? Is the benchmark intended to evaluate different LLMs in these two phases (schema extraction and execution)? Or is the benchmark for evaluating new LLM scaffolding/planning methods? Or is the benchmark for evaluating new LLM serving pipelines?
>
> As stated in the introduction, ParallelPrompt is a benchmark that enables both method and system evaluation. Specifically, it enables:
>
> 1. **Schema extraction research**, by evaluating how well models or prompting strategies identify parallelizable structure;
> 2. **Decomposition execution strategies**, through latency-quality tradeoff evaluation;
> 3. **Model benchmarking**, by assessing how different LLMs perform under parallel vs. serial execution across task categories.
>
> The task-agnostic schema format and modular infrastructure are designed to support any structure-aware execution pipeline, providing a standardized testbed for structure-aware LLM execution research.
>
>
> >> As a paper introducing a new benchmark I would expect evaluating multiple systems on this benchmark. Currently this is deferred to [Appendix I] which is quite confusing.
>
> We clarify that **multiple systems and methods are evaluated**, with their limitations discussed in the main text and Appendix I.
>
> - Section 2 introduces real-world decomposition methods (Skeleton-of-Thought, Tree-of-Problems).
>
> - Section 4.1 and 4.2 discusses briefly their evaluation and analyzes their failure points on our benchmark.
> - Appendix I provides a detailed dive and comparisons with the baseline execution strategies.
>
> We follow standard benchmark-paper structure, with our Eval section emphasizing core evaluation findings across critical dimensions (latency, scalability analysis, multiple systems discussion, quality-speedup tradeoffs, eval failure modes, practical implications) in the space-constrained main text while presenting full method comparisons in the appendix for transparency and completeness.
>
> >> How good is Claude-3.5 doing in Phase 1: schema extraction? Line 306 briefly mentioned manual inspection but more details is needed
>
> Section 3.2 and Appendix B detail our model comparison and validation strategy. While we didn't discuss this ablation in the paper, we tested Claude-3.7 Sonnet (too powerful and over-extracted), GPT-4o-mini (too weak and under-performed), and open-source models via Together.ai API. Claude 3.5 was chosen for optimal tradeoff of precision and coverage. If you deem this info useful, we can certainly add them for the camera-ready version.
>
> As stated in Section 5.2 (see paragraph 3), we conducted “~5,500 manual inspections” across our three-tier validation framework. Detailed quality metrics and failure analysis are provided in Appendices B and D.4.
>
>
> >> What did you do to balance precision and recall? How is precision and recall defined here?
>
> Thank you for flagging this ambiguity. In our context:
>
> - **Precision** is the fraction of *validated* prompts that can be executed in parallel without semantic degradation.
> - **Recall** is the fraction of *all inherently parallelizable prompts* that our pipeline successfully identifies and validates.
>
> Our pipeline (Section 3.2) explicitly **prioritizes precision**, using strict schema constraints and symbolic checks, and we agree that recall is currently conservative, and can improve with better schema extractors. We will clarify this distinction in the paper.
>
>
> >> I'm confused whether this section 3 is creating ground-truth annotation for your benchmark, or is it part of your proposed method to the benchmark?
>
> Section 3 describes our benchmark curation pipeline, not a proposed method for benchmark. The schema extraction and validation steps define the dataset annotations that other researchers can use to test their own decomposition methods or execution strategies.
>
>
> >> Line 116: How do you decide the seven canonical categories?
>
> Section 3.1 notes these categories as common parallelizable patterns. To elaborate: they emerged from painstaking manual curation looking through 10,000+ prompts from LMSYS-Chat-1M at the earliest phase of this work’s development. Through this initial analysis, we discovered recurring parallelizable patterns that motivated our schema definition, which then helped us identify the remaining canonical categories (since we now knew the markers to look out for) before turning to automated extraction with Claude for the final dataset curation – we can make this clearer in our revision. Detailed taxonomy is provided in Appendix C.
>
>
> >> Line 208: I am a little confused at the normalized speed up metric. What does 'parallel execution often produces more verbose and detailed outputs' mean?
>
> As explained in Appendix G.1, LLMs processing multiple items sequentially often truncate each individual answer, due to internal length ``budgeting” behavior. For example, asking for 1 story might yield 500 words, but asking for 10 stories results in ~50 words each as the model tries to fit everything into roughly the same total response length. This length delusion effect wrt quality degradation has been systematically documented [1,2,4], and [2,3] shows that this effect is mathematically predictable and scales with the number of items.
>
> Parallel execution avoids this by allocating full context to each subtask. Our normalized speedup metric adjusts for this by reporting **efficiency per output token**, rather than raw wall time, allowing a fair comparison of actual processing efficiency.
>
>
> >> I wonder how this work fits in to the literature of LLM planning/agents? Your discussion on task-adaptive execution sounds very relevant. I saw that DSPy and LangChain were briefly mentioned in the related work section. Would any of these serve as reasonable baselines?
>
> As discussed in Section 2, frameworks like DSPy, SGLang, and LangChain “enable parallel execution of downstream tasks generated from a single task, but also do not aim to discover execution structure within user queries.” These systems focus on orchestrating multiple calls for complex workflows, while our work identifies latent parallelism within individual user prompts.
>
> The connection to planning/agents is interesting; future work could explore how our identified parallelizable structures could inform more sophisticated agent decomposition strategies. However, these frameworks aren’t direct baselines since they solve a different problem (workflow orchestration vs. intra-query structure discovery).
>
>
> >> I wonder what the authors think about this paper (https://arxiv.org/abs/2301.08721) which seems to be advocating for the opposite of decomposing parallel prompts. Do you think the two paper are conflicting with each other or could you discuss some of the discrepancies?
>
> Thank you for this important reference. While batch prompting [2] and our approach may appear contradictory at first glance, they actually address different optimization targets and could be complementary:
>
> **Different starting points:** [2] starts with N separate, independent prompts from different users and combines them into single API calls. ParallelPrompt starts with 1 complex prompt that internally contains parallel structure and decomposes it into subtasks.
>
> **Different Optimization Targets:** [2] optimizes inter-query efficiency (across multiple requests) while our work optimizes intra-query efficiency (within a single complex request). The approaches operate at different granularities of the optimization pipeline.
>
> In fact, the approaches could potentially work together: (1) detect if a complex prompt has parallel structure, (2) decompose into subtasks if beneficial, (3) group subtasks into efficient batches for API calls. For example, “Translate these 20 sentences” could be decomposed into 20 translation subtasks, then batched into 4 API calls with 5 translations each. Rather than conflicting, both works suggest we need adaptive strategies that can intelligently choose between batching, decomposition, or hybrid approaches based on the specific prompt and context.
>
>
> We appreciate your thorough questions, which have identified several areas where our presentation could be clearer and more complete.
>
> ---
>
> [1] Son, G., Baek, S., Nam, S., Jeong, I., & Kim, S. *Multi-Task Inference: Can Large Language Models Follow Multiple Instructions at Once?* ACL 2024.
>
> [2] Cheng, Z., Kasai, J., & Yu, T. *Batch Prompting: Efficient Inference with Large Language Model APIs.* In EMNLP (Industry Track) 2023.
>
> [3] Zheng, Z., Ren, X., Xue, F., Luo, Y., Jiang, X., & You, Y. *Response length perception and sequence scheduling*: An llm-empowered llm inference pipeline. NeurIPS 2023.
>
> [4] Guldogan, O., Kunde, J., Lee, K., & Pedarsani, R. (2024). *Multi-bin batching for increasing LLM inference throughput.* arXiv preprint arXiv:2412.

---

> > ### Comment · Reviewer_Zyke · 2025-08-05
> >
> > Thank you for the response and discussion. I'm happy to increase my rating since my concerns are mostly addressed.
> >
> > * The human validation on LLM-annotated data is quite important so I would suggest summarizing your findings in Appendix B and D.4 in an accessible way in the main paper.
> > * Many parts of the paper requires reading extra context from the appendix, which makes the paper a little hard to follow. I know it's hard to compress the paper into the page limit but I hope this could be further improved.

---

> > > ### Author Response · Authors · 2025-08-08
> > >
> > > Thank you for increasing your rating and for the detailed feedback! We look forward to making the improvements of your not-yet-addressed suggestions in the camera-ready version.

---

### Official Review · Reviewer_HJzY · 2025-07-02

**Rating:** 4
**Confidence:** 4

**Summary:**

ParallelPrompt proposes that some of the user prompts could be executed in parallel, and they curate a benchmark to demonstrate parallel execution.

**Additional Feedback:**

I think overall, the benchmark construction and structure are okay, but the actual implementation and extendability of the benchmark have problems. Moreover, the problem setting (executing user prompts in parallel speeds up the overall serving) is not sufficiently justified by the dataset collection. Only 10% of the data (37K / 385K) is marked as parallelizable, which is a quite small portion of the total requests. But intuitively, I think we can find more data that can be executed in parallel and realistically, such as retrieval tasks in an agentic AI scenario.

**Dataset Code Accessibility:**

Yes

**Dataset Code Comments:**

- I think the dataset code structure is not in good condition; therefore, I strongly suggest that the authors organize their README and dataset code structure. It looks pretty hard to understand which module does what job.
- Furthermore, the programming language in this code base is too variant, which makes it hard for future researchers to modify the code base for creative experiments. I think the modification-ability of benchmark and dataset code is pretty important, so I strongly suggest that authors use at least a unified single widely used program language, Python. (By the way, I like to use JS and C++ too, so personally I think using those languages is okay too. But I am not sure other researchers think about this, because usually everyone uses Python.)

**Ethical Considerations:**

No, there are no or only very minor ethics concerns

**Final Justification:**

I think this paper is acceptable

**Limitations Weaknesses:**

- The dataset is only curated from just two datasets of the multi-turn dataset. The total number of data examples looks sufficient to benchmark the future proposed frameworks, but I am worried that only 10% of examples are marked as parallelizable. This seems to be a pretty low number for justifying that prompt parallelism is important to speed up overall serving throughput. I think authors can find a better dataset that contains some parallelly executable prompts, such as multi-agent prompts.

**Strengths Contributions:**

Parallel execution of prompt sounds pretty important in many tasks in the real world, such as retrieval tasks. This paper points out where we can apply parallel executions and how to measure the performance gain with the total dataset.

---

> ### Author Rebuttal · Authors · 2025-07-30
>
> We thank the reviewer for acknowledging that `parallel execution of prompt sounds pretty important in many tasks in the real world` and for recognizing our contributions in highlighting `where we can apply parallel executions and how to measure the performance gain with the total dataset`. We address your concerns in detail below:
>
> >> The dataset is only curated from just two datasets of the multi-turn dataset. The total number of data examples looks sufficient to benchmark the future proposed frameworks, but I am worried that only 10% of examples are marked as parallelizable. This seems to be a pretty low number for justifying that prompt parallelism is important to speed up overall serving throughput.
>
> We respectfully believe that the 10.3% schema validation rate is being misinterpreted as a limitation. In practice, this figure reflects a **high-precision, conservative estimate** of structurally decomposable prompts under real-world conditions, not an upper bound of potential parallelism.
>
> The practical significance is substantial: even if only 10% of prompts are parallelizable, this translates to **millions of opportunities per month** in high-traffic deployments (e.g., ChatGPT’s >1B monthly queries). Unlike inter-query batching which depends on concurrent load, intra-query decomposition offers consistent acceleration opportunities, delivering up to 5x speedups without sacrificing throughput or batching opportunities elsewhere (see Table 1).
>
> For context, even widely-adopted optimizations like speculative decoding and KV-caching only benefit certain query types, yet are considered valuable production techniques. Our 10% baseline parallelism is consistently available within individual queries and complements existing optimizations rather than competing with them.
>
> Further, we clarify that this 10% should be seen as a **lower bound**. Our validation pipeline (App. B-D) prioritizes semantic precision and structural reproducibility over recall. As schema extraction improves via better prompting, multi-lingual adaptation, and cross-example generalization, we expect coverage to increase. Our benchmark is designed as a reproducible foundation for measuring and tracking such progress.
>
>
> >> I think authors can find a better dataset that contains some parallelly executable prompts, such as multi-agent prompts.
>
> We appreciate the suggestion. We deliberately curated ParallelPrompt from LMSYS-Chat-1M and WildChat-1M because they represent the largest publicly available collections of **natural, unconstrained, real user-LLM interactions**. While multi-agent tasks may show higher rates of decomposability, they are more niche, consisting of scripted scenarios lacking the organic variability seen in open-ended user prompts.
>
> Our benchmark aims to establish a reliable lower-bound estimate of parallelism in general-purpose LLM usage, not artificially maximize decomposable prompts. That said, we welcome future benchmarks focused on multi-agent or retrieval heavy workflows, and view these as complementary.
>
> >> I think the dataset code structure is not in good condition; therefore, I strongly suggest that the authors organize their README and dataset code structure. It looks pretty hard to understand which module does what job.
>
> We appreciate reviewer’s feedback about code clarity. We agree that a benchmark’s usefulness depends on accessibility and ease of extension. . Following your suggestion, we have:
>
> - refactored the codebase README with clearer entry points for the each module (e.g., schema generation, validation. execution).
> - added inline docstrings and usage examples to clarify pipeline stages.
> - ensure the documentation links modular tasks (data loading, schema extraction, parallel execution to the corresponding scripts.
>
> These updates are already reflected in the public repo, and we welcome further usability feedback.
>
>
> >> Furthermore, the programming language in this code base is too variant, which makes it hard for future researchers to modify the code base for creative experiments.
>
> We used C++ for the backend because Python was not able to handle multiple network connections in a performant way and C++ was necessary to get the parallelization to work. The multi-language architecture (Python curation, C++ execution) reflects real-world deployment patterns where curation and serving often use different technologies for performance reasons.
>
> However, we take your feedback seriously and will add in the future a Python wrapper interface that allow researchers to work entirely in Python while leveraging the high-performance C++ backend transparently.

---

> > ### Comment · Reviewer_HJzY · 2025-08-05
> >
> > Thank you for your insightful response!
> >
> > > Further, we clarify that this 10% should be seen as a lower bound.
> >
> > This is a new aspect that I could not see. I agree with the authors that the importance of the problem is high.
> >
> > > I think authors can find a better dataset that contains some parallelly executable prompts, such as multi-agent prompts.
> >
> > As long as my first concern was resolved, this should be resolved for the same reason.
> >
> > > We appreciate the reviewer’s feedback about code clarity.
> >
> > I appreciated seeing the updated codebase. However, I have a question: Can I test my custom OpenAI API compatible servers, such as SGlang or vLLM? In the readme, I cannot see such a feature.
> >
> > > C++ was necessary to get the parallelization to work
> >
> > Yes, I understand it, but I think it is still hard to follow for non-engineering-skilled researchers. I think a Python wrapper sounds like a good option to do it, but it still limits further modification ability. I hope the author can find a better way to construct the framework with easier language and framework (maybe Node.js is better?)
> >
> > -------
> >
> > Overall, I am happy to see their work to improve the code structure. Since I think the paper looks pretty nice, and the codebase is ready to launch, I will vote to accept by raising the score from 3 to 4.

---

> > > ### Author Response · Authors · 2025-08-08
> > > **Follow-up Response**
> > >
> > > Thank you for upgrading your rating and for the constructive feedback throughout the review process!
> > >
> > > **Re custom OpenAI-compatible servers:** We've now added support for SGLang, vLLM, and other OpenAI-compatible servers using the standard OPENAI_API_BASE environment variable. Updated usage examples are in the README.

---

### Official Review · Reviewer_qkE1 · 2025-07-23

**Rating:** 5
**Confidence:** 3

**Summary:**

This paper introduces the a public benchmark that explicitly targets intra‑query semantic parallelism in LLM prompts. The authors curated ~ 37 k naturally occurring user prompts from LMSYS‑Chat‑1M and WildChat‑1M and annotate each with a structured schema that shows how the prompt can be decomposed into independent subtasks. Also,  an automated curation/validation pipeline plus an evaluation suite that contrasts serial and parallel execution in terms of latency, structural fidelity, and semantic quality, are released. It also collected some meaningful empirical observation. With GPT‑4‑1106‑preview as the tested model, the benchmark demonstrates up to > 5 × speed‑ups on tasks such as reading comprehension, and shows that some creative writing tasks suffer more regression with parallel processing.

**Dataset Code Accessibility:**

Yes

**Dataset Code Comments:**

All codes can be found in the given repo.

**Ethical Considerations:**

No, there are no or only very minor ethics concerns

**Final Justification:**

The remaining issue is promised to be solved in the camera-ready version. Considering the novelty of this work, "accept" will be appreciated.

**Limitations Weaknesses:**

1: The imbalance of language might be a concern. I understand that may originates from the bias of current LLM as the data is from real world traffic. But a more balanced version will be preferred as the behavior of LLMs is different for different language, which is also found in the paper.

2: A Single‑model is tested. The execution uses only GPT‑4‑1106‑preview, which limits its generalizability to open‑source or smaller models. Also, as a remotely hosted model, it's not proper to discuss about efficiency based on it.

**Strengths Contributions:**

1: It is a novel, as far as I know it's the first large‑scale benchmark for parallel structure process in prompts. Also, it is grounded in real user queries rather than synthetic tasks.

2: The quality control part is sophisticated. The three‑level pipeline combines LLM extraction with symbolic checks, ensuring each prompt is paired with a reproducible template/context/data schema for downstream use

3: The empirical results also provide insights to the community. The reference runs show 2.5 ×–5.7 × raw speed‑ups (3.4 × end‑to‑end for repeated generation) while preserving >90 % quality on factual tasks, illustrating the practical value of structure‑aware execution.

---

> ### Author Rebuttal · Authors · 2025-07-30
>
> We thank the reviewer for recognizing the novelty of our work and that is `grounded in real user queries rather than synthetic tasks.` We appreciate your assessment that our `quality control part is sophisticated` and that the empirical results demonstrate `practical value of structure-aware execution.` We address the concerns below:
>
> >> The imbalance of language might be a concern. I understand that may originates from the bias of current LLM as the data is from real world traffic. But a more balanced version will be preferred as the behavior of LLMs is different for different language, which is also found in the paper.
>
> The 84% English distribution reflects authentic user interaction patterns from two of the largest public LLM conversation datasets available and provides valuable research opportunities by exposing (for the first time, to the best of our knowledge) genuine multilingual challenges that need addressing.
>
> Rather than presenting an artificially balanced dataset that would obscure these real-world deployment challenges, ParallelPrompt provides the first standardized testbed for developing language-aware schema extraction methods, while also providing detailed failed pattern analysis (Appendix E) to guide future improvements.
>
>
> >> A Single‑model is tested. The execution uses only GPT‑4‑1106‑preview, which limits its generalizability to open‑source or smaller models. Also, as a remotely hosted model, it's not proper to discuss about efficiency based on it.
>
> While our reference execution uses GPT-4-1106-preview, the benchmark is explicitly designed to be model-agnostic, and gains are expected to generalize beyond proprietary models. The schema format (Appendix A.2) and C++ suite (Appendix F) are designed for plug-and-play execution with any API-compatible model.
>
> Also, we note that the focus of our benchmark is to assess *decomposition strategies*, not specific model internals. That said, while we didn’t include such ablations in the paper, we extensively experimented with model variants throughout the entire benchmark pipeline. For example, for the schema extraction phase, we tested Claude-3.7 Sonnet (too powerful and over-extracted), GPT-4o-mini (too weak and under-performed), and open-source models via Together.ai API. Claude 3.5 Haiku was chosen for optimal tradeoff of precision and coverage. If you deem this info useful, we can certainly add them for the camera-ready version.
>
> Finally, we note that our reported latency gains (e.g., 3.41x for Repeated Generation, Table 1) are conservative due to API latency overheads, but they still represent genuine efficiency gains that can translate directly to production cost savings. We agree that these benefits would be amplified with local model deployment where network latency is eliminated.

---

> > ### Comment · Reviewer_qkE1 · 2025-08-08
> >
> > Thank you for the response. "If you deem this info useful, we can certainly add them for the camera-ready version." such info would be appreciated in the final version.
> >
> > I will keep my score as it is already positive.

---

### Decision · Program_Chairs · 2025-09-18

**Decision:**

Accept (poster)

**Comment:**

The reviewers unanimously agree that the paper makes significant contributions the benchmarking effort of parallel processed LLM prompts. It is by far the first such dataset and can potentially offer useful value to the field.